# *C. elegans* monitor energy status via the AMPK pathway to trigger innate immune responses against bacterial pathogens

Shouyong Ju[1,3], Hanqiao Chen [1,3], Shaoying Wang[1], Jian Lin[1], Yanli Ma[1], Raffi V. Aroian[2], Donghai Peng [1,4✉] & Ming Sun[1,4✉]

Pathogen recognition and the triggering of host innate immune system are critical to understanding pathogen-host interaction. Cellular surveillance systems have been identified as an important strategy for the identification of microbial infection. In the present study, using *Bacillus thuringiensis-Caenorhabditis elegans* as a model, we found an approach for surveillance systems to sense pathogens. We report that *Bacillus thuringiensis* Cry5Ba, a typical pore-forming toxin, caused mitochondrial damage and energy imbalance by triggering potassium ion leakage, instead of directly targeting mitochondria. Interestingly, we find *C. elegans* can monitor intracellular energy status to trigger innate immune responses via AMP-activated protein kinase (AMPK), secreting multiple effectors to defend against pathogenic attacks. Our study indicates that the imbalance of energy status is a prevalent side effect of pathogen infection. Furthermore, the AMPK-dependent surveillance system may serve as a practicable strategy for the host to recognize and defense against pathogens.

[1] State Key Laboratory of Agricultural Microbiology, Hubei Hongshan Laboratory, National Engineering Research Center of Microbial Pesticides, Huazhong Agricultural University, Wuhan 430070, China. [2] Program in Molecular Medicine, University of Massachusetts Chan Medical School Worcester, Worcester, MA 01605-2377, USA. [3] These authors contributed equally: Shouyong Ju, Hanqiao Chen. [4] These authors jointly supervised this work: Donghai Peng, Ming Sun. ✉email: donghaipeng@mail.hzau.edu.cn; m98sun@mail.hzau.edu.cn

Animals encounter diverse pathogens in the natural environment, and they have evolved different defense responses against pathogens for survival. The primary challenge for the host is how to sense the pathogens and trigger defense responses. The innate immune system is a universal and evolutionarily ancient part of such host defense response[1].

It is generally accepted that hosts can discriminate pathogenic bacteria from non-pathogenic bacteria in multiple ways. First, the host can recognize microbe-associated molecular patterns (MAMPs) or endogenous danger-associated molecular patterns (DAMPs) by pattern-recognition receptors (PRRs) to induce immune signaling pathways[2]. This is also known as pattern-triggered immunity (PTI), which is the most traditional way for hosts to identify pathogens. It is well-known that MAMPs are highly conserved and common among microbes[3,4]. DAMPs are endogenous molecules secreted by damaged cells[5], implying that they are not the premier immune-activating factor. However, the PRR ligands are not unique to pathogenic bacteria, but also found in the non-pathogenic bacteria[6], making it hard to simply identify the pathogen. Besides, hosts can sense certain virulence factors or damage to discriminate pathogens, a process known as effector-triggered immunity (ETI)[7–9]. The ETI responses largely explain how the host cells identify pathogens from environmental microorganisms. Most of the ETI responses still depend on PPRs, while mounting evidence shows that *Caenorhabditis elegans* could sense pathogens without activating PPRs through "surveillance immunity"[10,11], which is the important part of ETI response[12,13]. Therefore, PTI and ETI responses largely explain the mechanism of pathogen recognition.

Recently, there are increasing studies concerned how the host could indirectly detect pathogens through non-PPRs-related cellular surveillance responses. In *C. elegans*, host cellular surveillance systems could monitor the core cellular activities disorder caused by pathogens, triggering innate immune responses through non-PPRs-related ways. The integrity of nucleolus[14] or DNA[15], inhibits the transcription or translation processes[13,16], and disruption of the mitochondrial protein folding environment[17,18] can be sensed by host cellular surveillance systems to initiate innate immune defenses. Consequently, host cellular surveillance response is an important extended strategy under ETI responses during pathogen recognition process. Despite the continuous enrichment of the immune surveillance systems, the upstream specific signaling molecules and pathways representing the changes of cell homeostasis to induce the innate immune defenses remain largely unknown.

By far, the most well-known reported cellular surveillance systems contain the ribosome, proteasome, and mitochondria[19]. Numbers of work showed that the mitochondria could be attacked by toxins produced from pathogens. Toxins from many bacterial pathogens like vacuolating toxin (VacA) in *Helicobacter pylori*[20], alpha-toxin in *Clostridium perfringens*[21], and leucocidin in *Staphylococcus aureus*[22] can target and damage the mitochondria, resulting in mitochondrial dysfunctional. Furthermore, *C. elegans* can activate mitochondrial UPR(UPR^mt) response by the transcription factor ATFS, which eventually engages the host innate immune defenses to defense pathogens[17,18]. These studies indicate that the mitochondrial surveillance system is an effective means for host to monitor pathogen infection.

The mitochondria are vital biosynthetic and bioenergetic organelles that play a key role in cell homeostasis[23,24], Which can be disrupted by internal and external "stressors"[25]. Mitochondria can sense and respond to several kinds of stressors, including environmental change, genetic mutation, endogenous molecules and even to pathogens[26]. Several pathogen toxins can act as stressors to directly target mitochondrial or related proteins. However, mitochondria sense and respond to cell endogenous stressors caused by pathogens is remains to be studied.

The *C. elegans* has been developed as a powerful model to study innate immune responses. Numerous studies supported that *C. elegans* can monitor core cellular physiology activities to detect pathogen infections[13,16,19]. Moreover, no exact PPRs have been unambiguously defined in *C. elegans*[27]. These characteristics make *C. elegans* an ideal model to investigate how surveillance systems sense pathogens through non-PPR patterns. *Bacillus thuringiensis* (Bt) is an obligate and opportunist pathogen of insects and worms, which produces insecticidal or nematicidal crystal toxin (Cry) during sporulation and has been used as a leading bio-insecticide to control various insect pests[28]. Here, we used a nematicidal Bt strain BMB171/Cry5Ba[29] and *C. elegans* to research the detailed mechanisms about how the cell surveillance systems sense pathogens and activate the innate immune responses. We showed that Bt infection causes a severe cellular energy imbalance by altering of AMP/ATP ratio. Instead of directly targeting mitochondria, the energy imbalance was due to the mitochondrial damage caused by leakage of potassium resulting from the Cry toxin. The Cry toxin-mediated energy imbalance triggered the innate immune response via a cellular energy sensor AMPK in *C. elegans*. By secreting multiple effectors, the DAF-16 and p38-MAPK-dependent signaling pathways were activated to resist pathogen infection. Our work reveals mitochondrial surveillance systems can discriminate pathogens from the non-pathogenic bacteria via cell energy sensor AMPK, and then trigger innate immune to defense against bacteria pathogen infection in *C. elegans*, which provide insights into host-pathogen interactions.

## Results

**B. thuringiensis infection leads to cellular energy imbalance in C. elegans.** To investigate the intracellular physiological changes of *C. elegans* after pathogenic Bt infection, we conducted a transcriptome analysis of *C. elegans* after infection by the nematicidal Bt strain BMB171/Cry5Ba, an acrystalliferous Bt mutant BMB171 transformed with toxin gene *cry5Ba* on the shuttle vector pHT304[29]. As a control, we compared the transcriptome to a non-nematicidal Bt strain BMB171/pHT304, BMB171 transformed with the empty vector pHT304. Enrichment pathway analyses by KOBAS highlighted several pathways strongly affected by the infection of nematicidal Bt strain (Fig. 1a and Supplementary Data 2, 3). Interestingly, we found the metabolic-related pathway was most strongly affected. So we speculated whether the energy metabolic pathway could be impacted by Bt infection. To confirm this, we measured the concentrations of AMP and ATP by LC-MS[30] when wild-type *C. elegans* N2 fed with BMB171/Cry5Ba, BMB171/pHT304, and the standard food strain *E. coli* OP50. The results showed that the AMP/ATP ratio had no significant difference after *C. elegans* N2 fed with BMB171/pHT304, but significantly increased after fed with BMB171/Cry5Ba, compared with that of fed with *E. coli* OP50 (Fig. 1b). However, the AMP/ATP ratio showed no significant difference when the Cry5Ba-receptor mutant *bre-5(ye17)* worms were fed with any strains, which is resistant to Cry5Ba exposure (Fig. 1b).

Furthermore, we investigated whether other nematicidal Bt can lead to cell energy change, such as BMB171/Cry5Ca[29], BMB171/Cry21Aa[31], BMB171/Cry6Aa[32], and non-nematicidal Bt BMB171/Cry1Ac[33]. These former two strains produce Cry5-like three-domain (3D) group nematicidal Cry proteins Cry5Ca[29] and Cry21Aa[31], respectively, while BMB171/Cry6Aa produces a non-3D group nematicidal Cry toxin Cry6Aa[34]. We found that all nematicidal Bt strains can cause significant energy imbalance of *C. elegans* compared to non-nematicidal Bt strains (Supplementary Fig. 1). Taking together, we demonstrated that nematicidal Bt

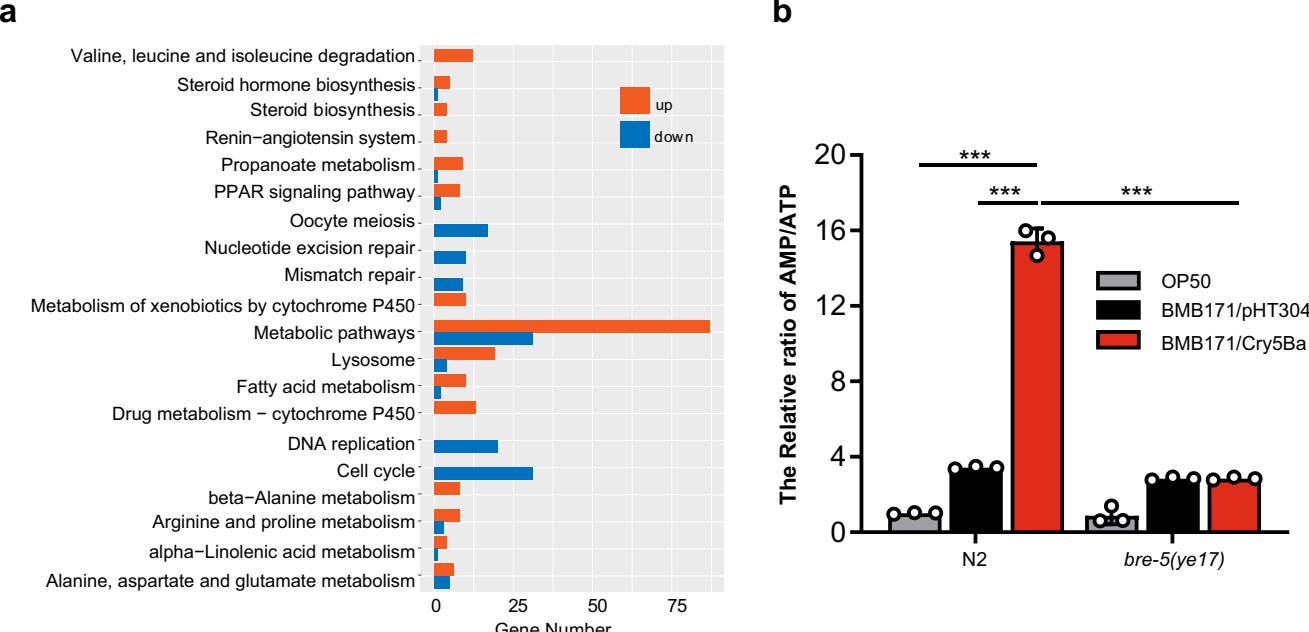

**Fig. 1 Nematicidal Bt infection leads to energy imbalance in *C. elegans*. a** An Enrichment Pathway analysis of RNA-Seq data using KOBAS showing the top 20 pathways influenced by nematicidal Bt infection. **b** Analysis of the AMP/ATP ratio by LC/MS. The wild-type N2 and *bre-5(ye17)* worms were fed with OP50, BMB171/pHT304, and BMB171/Cry5Ba for 4 h. $N = 3$ independent experiments. Data points represent the mean values of three independent replicates, error bars denote the SD. The *p*-value was determined by a Student's *t*-test (Welch's correction for unequal variances), ***$p < 0.001$.

infection triggers a cellular energy imbalance of *C. elegans*, which was mainly attributed to nematicidal toxins.

***B. thuringiensis* infection leads to mitochondria damage.** Mitochondria produce most of the ATP through oxidative phosphorylation and the tricarboxylic acid cycle and play a vital role in cell metabolism[35]. Mitochondria are constantly fusing and dividing, which is essential for maintaining mitochondrial respiration and homeostasis, even in cell death[36,37]. It has been proven that pore-forming toxins (PFTs) from pathogens can cause serious mitochondrial disruption in *C. elegans*[17,18,38]. The mitochondria fragmentation (MF) phenomenon can lead to bioenergetics defects, resulting in an imbalance of cell energy[39]. To assess how nematicidal Bt causes cellular energy imbalance, we examined several physiological and biochemical aspects of mitochondrial damage including MF, mitochondrial membrane potential ($\Delta\Psi_m$), and mitochondrial DNA (mtDNA) content. We used the transgene worm SJ4143($zcIs17[P_{ges-1}$::GFP$^{mt}]$) as an MF reporter to assess mitochondrial morphology, which stably expresses GFP in mitochondria matrix of intestinal cells[40]. Most worms showed MF when fed with nematicidal Bt BMB171/Cry5Ba. However, when worms were fed with BMB171/pHT304 or OP50, the mitochondria morphology of most worms was tubular, with around 15% of worms showing MF (Fig. 2a, b). These results showed that nematicidal Bt infection leads to increases in MF. Next, we examined whether BMB171/Cry5Ba infection can cause changes in $\Delta\Psi_m$ and mtDNA content. Using wild-type N2 worms, we found BMB171/Cry5Ba infection can result in a considerable decrease in $\Delta\Psi_m$ and mtDNA content compared with non-nematicidal Bt BMB171/pHT304 (Fig. 2c, d and Supplementary Fig. 2). These results prove that nematicidal Bt can cause severe mitochondrial functional damage.

When the Cry5Ba-receptor-related gene *bre-5* was silenced by RNA interference (RNAi) in the transgene worm strain SJ4143, only less than one-fifth of the worms showed MF phenomenon after BMB171/Cry5Ba treatment (Fig. 2a, b), indicating that the

Cry protein toxin Cry5Ba was the key factor to cause the MF phenomenon during Bt infection. However, when Cry5Ba-receptor null mutant *bre-5(ye17)* worms were fed with BMB171/Cry5Ba in the same condition, the $\Delta\Psi_m$ and mtDNA were not changed significantly (Fig. 2c, d). We conclude that Cry5Ba toxin is the key factor for Bt to manipulate host cell mitochondria and cause serious mitochondrial dysfunction.

Mdivi-1 is an efficient inhibitor that attenuates mitochondrial division by inhibiting the mitochondrial division dynamin; it also suppresses mitochondrial outer membrane permeabilization[41]. To assess the relationship between cell energy imbalance and mitochondrial dysfunction after Bt infection, we investigated whether mitochondrial division inhibitor mdivi-1 can reduce mitochondria damage. We observed that mdivi-1 can efficiently reduce Cry5B-mediated MF and restore the AMP/ATP ratio, as well as the mtDNA/nDNA ratio, to normal levels (Fig. 2a, b, d and e). However, we found mdivi-1 cannot recover the reduction of $\Delta\Psi_m$ (Fig. 2c). In general, these results indicate that mitochondrial damage is responsible for the increase in intracellular AMP/ATP.

To assess whether other nematicidal Bt strains can cause MF phenomenon, we fed transgene worms SJ4143 with the other four nematicidal Bt as described above. The transgene worms SJ4143 fed with nematicidal BMB171/Cry5Ca, BMB171/Cry6Aa, and BMB171/Cry21Aa exhibited the fragmented mitochondrial morphology in varying degrees. However, the worms fed with BMB171/Cry1Ac showed no significant MF phenomenon (Supplementary Figs. 3, 4). The results showed that toxins from nematicidal Bt are capable of causing mitochondrial damage in *C. elegans*.

***B. thuringiensis*-induced mitochondrial damage is the result of intracellular potassium leakage.** The Cry5Ba toxin is a typical PFT that can form pores in the cell membrane and disrupt plasma membrane permeability[42,43]. To explain the mechanism of Cry5Ba caused mitochondrial damage, we determine the direct stressors of mitochondria in response to Cry5Ba. We first check

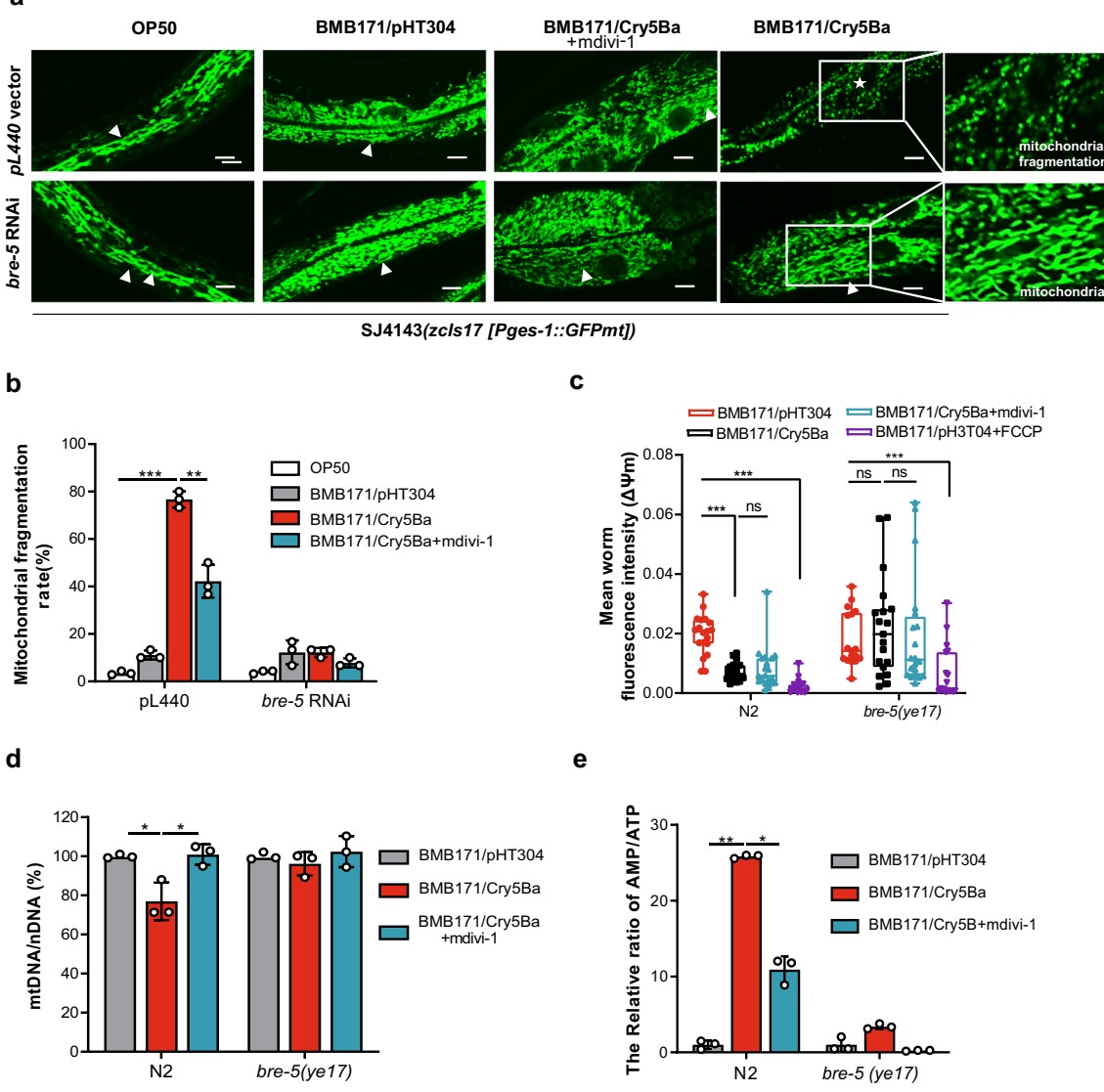

**Fig. 2 Bt infection leads to mitochondria damage. a** Observations of the mitochondrial morphology of transgenic *C. elegans* SJ4143(zcIs17 [P$_{ges-1}$::GFP$^{mt}$]) fed without or with *bre-5* RNAi. The images depicted are representative images of worms treated by each strain (triangle: mitochondria in the fusion state, showing tubular shape; star: mitochondria in the split state, showing punctate shape which represents mitochondria fragmentation (MF). The scan bars represent 20 μm. $N = 3$ independent experiments. **b** Transgenic worms were treated with each Bt strain after being subjected to *bre-5* RNAi (resistant to toxin exposure) or empty vector. Count the percentage of worms with the mitochondria fragmentation phenotype after treatment in each condition. $N = 3$ independent experiments containing at least 50 worms each. **c** Analysis of the mitochondrial membrane potential (ΔΨm) by fluorescent dye stetramethyl rhodamine ethyl ester under each treatment for 4 h. The level of ΔΨm was determined by the fluorescence intensity of each worm. $N = 3$ independent experiments containing at least 20 worms. **d** The relative ratio of mtDNA /nDNA was investigated by real-time PCR. $N = 3$ independent experiments. **e** The relative AMP/ATP ratio was measured in each condition. In each experiment, worms were treated for 4 h. $N = 3$ independent experiments. FCCP: a protonophore can decrease mitochondrial membrane potential; mdivi-1: an efficient inhibitor that attenuates mitochondrial division. Data points represent the mean values of three independent replicates, error bars denote the SD in (**b**), (**d**), and (**e**). The whiskers represent the data range (minimum–maximum), and the box shows the 75th percentile, 25th percentile, and median in (**c**). The *p*-value was determined by a Student's *t*-test (Welch's correction for unequal variances), \*\*\*$p < 0.001$, \*\*$p < 0.01$, \*$p < 0.05$.

whether Cry5Ba can directly target mitochondrial. By feeding SJ4143 worms with rhodamine-labeled Cry5Ba and using the endocytosis-labeled worm RT311[P$_{vha-6}$::GFP::RAB-11] as a control, our results showed Cry5Ba is translocated into epithelial cells as shown by white spots, but does not colocalize with mitochondria (Fig. 3a). In addition to targeting mitochondria directly, it had been proved PFTs can lead to severe ion dysregulation, which has been demonstrated to be important for mitochondria[44,45]. So we assessed the cellular Ca$^{2+}$ and K$^+$ levels after Bt infection. We found that Bt infection caused a significant

decrease in potassium ion, but not calcium ion, concentration (Fig. 3b and Supplementary Fig. 6). In a K$^+$-free environment, the leakage of potassium is more serious. When the potassium concentration increased, the leakage of potassium could be alleviated. Conversely in *bre-5(ye17)* worms, potassium content did not change at any condition (Fig. 3b and Supplementary Fig. 5). Thus we can conclude that during the Bt infection, Cry5Ba caused intracellular potassium leakage.

To verify our hypothesis that potassium leakage is the direct reason for mitochondrial damage. We detected the MF

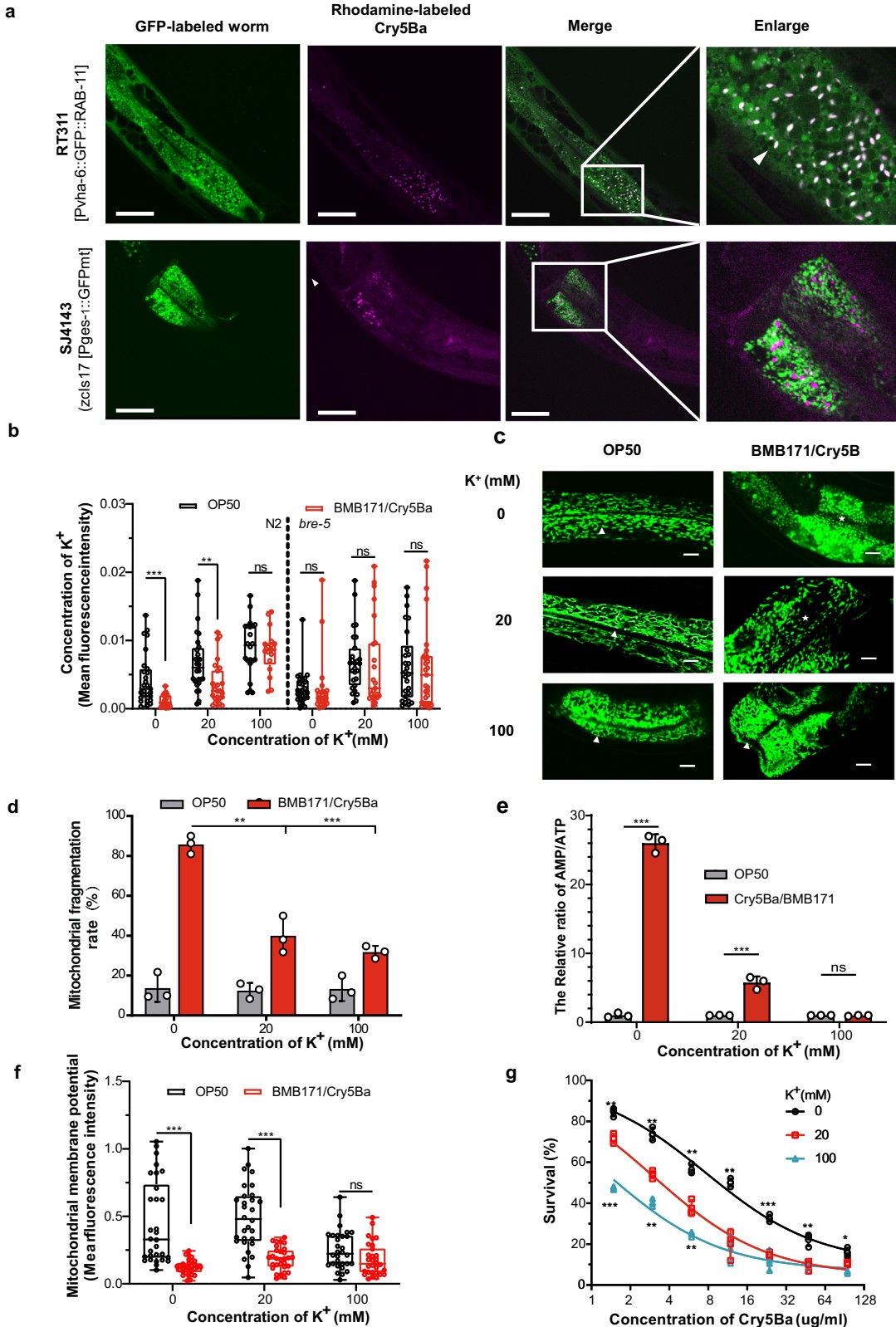

phenomenon during different potassium concentrations. We found during the Bt infection, the restoring of intracellular potassium can also reduce the MF phenotype, but serious potassium leakage caused a greater proportion of MF phenotypes (Fig. 3c, d). At the same time, the $\Delta\Psi_m$ could also return to normal level with potassium addition (Fig. 3f). As we found above, the MF phenotype

caused by Cry5Ba toxin is responsible for energy imbalance. Our results indeed showed the restoration of potassium concentration can restore the content of AMP/ATP (Fig. 3e). Therefore, we concluded that the leakage of potassium ions caused by Bt infection directly causes mitochondrial stress, which subsequently leads to mitochondrial damage and energy imbalance.

**Fig. 3 Intracellular potassium leakage caused mitochondrial damage in *C. elegans*. a** The fluorescent confocal micrograph of nematodes carrying a labeled endocytosis protein (RT311[Pvha-6::GFP::RAB-11]) and labeled mitochondria (SJ4143[zcIs17 [Pges-1::GFPmt]) fed rhodamine-labeled Cry5Ba protein. Cry5Ba (magenta) and endocytosis (green) were co-localized and displayed white dots (arrow), but not with mitochondria (green). Scale bar represents 10 μm. *N* = 3 independent experiments. **b** Quantitative the intracellular potassium concentrations by fluorescent dye ION Potassium Green-2 AM during *C. elegans* N2 and *bre-5(ye17)* treated with Cry5Ba/BMB171 at different potassium concentrations. *N* = 3 independent experiments containing at least 20 worms. **c** *C. elegans* SJ4143 was fed with OP50 or Cry5Ba/BMB171 under different potassium concentrations. The mitochondrial morphology was visible in representative images of worms treated by each strain. Scale bar represents 20 μm. *N* = 3 independent experiments. **d** Chart showing the proportion of worms that showed MF phenotype in each condition. *N* = 3 independent experiments containing at least 50 worms each. **e** Analysis of the mitochondrial membrane potential (ΔΨm)) in each condition under Cry5Ba/BMB171 treatment. *N* = 3 independent experiments containing at least 20 worms. **f** The relative AMP/ATP ratio was measured in different potassium environments with Cry5Ba/BMB171 treatment or not. *N* = 3 independent experiments **g** The survival assay of the N2 worms after BMB171/Cry5Ba infection at different potassium concentrations. *N* = 3 independent experiments. In experiment (**a**–**f**), worms were treated for 4 h. Data points represent the mean values of three independent replicates, error bars denote the SD in (**d**), (**e**), and (**g**). The whiskers represent the data from the maximum to the minimum, and the box shows the 75th percentile, 25th percentile, and median in (**b**) and (**f**). The *p*-value was determined by a Student's *t*-test (Welch's correction for unequal variances), \*\*\**p* < 0.001, \*\**p* < 0.01, \**p* < 0.05.

**Cell energy imbalance mediated by mitochondria damage activates the AMP-activated protein kinase**. The AMP-activated protein kinase (AMPK) is a sensor of energy status that maintains energy homeostasis and can be activated by a decrease in energy levels[46]. The synthesis and catabolism of ATP are largely regulated by AMPK[46]. AMPK is activated via phosphorylation of Thr172 on the α catalysis subunit (AAK-2 protein)[46], which has been reported to work as a candidate immunomodulator under Bt infection[47]. It is well-known that increasing AMP/ATP proportion is the classical way to activate the AMPK[46]. We observed that *aak-2* was significantly up-regulated and the AMP/ATP ratios were significantly increased when worms are infected by nematicidal Bt (Fig. 1b and Supplementary Fig. 7). Therefore, we speculated that nematicidal Bt infection may activate AMPK. To confirm this hypothesis, we performed western blotting to analyze the Thr172 phosphorylation of AAK-2 protein in worms. The results showed that the Thr172 of AAK-2 was phosphorylated when *C. elegans* fed with BMB171/Cry5Ba but not in the control strain BMB171/pHT304 and OP50 treatment (Fig. 4a, b). Furthermore, the Thr172 of AAK-2 protein was not phosphorylated when Cry5Ba-receptor null mutant *bre-5(ye17)* worms were fed with BMB171/Cry5Ba under the same conditions (Fig. 4a, b and Supplementary Figs. 8, 9). Inhibition of mitochondrial fragmentation using mdivi-1 could also significantly suppress the Thr172 phosphorylation of AAK-2 (Fig. 4a, b and Supplementary Figs. 8, 9).

Metformin has been reported to significantly increase intracellular AMP levels[48]. We first determined that it can significantly increase intracellular AMP/ATP levels (Supplementary Fig. 10), and further western blot analyses determined that it can significantly activate AMPK via AAK-2 Thr172 phosphorylation (Fig. 4a, b and Supplementary Figs. 8, 9). As a control, we activated AMPK by 50 μM of 5-Aminoimidazole-4-carboxamide 1-β-D-ribofuranoside (AICAR) (Fig. 4a, b), which is a typical activator stimulating AAK-2 kinase activity of AMPK via phosphorylation of Thr172 on α catalysis subunit[49]. Besides, we demonstrate that potassium leakage could directly lead to the activation of AMPK, restoring the leakage of potassium ions during Bt infection could suppress the Thr172 phosphorylation of AAK-2 (Fig. 4c, d). These results demonstrated that the AMPK of worms is activated by nematicidal Bt infection via the phosphorylation of the core subunit AAK-2.

To assess the relationship between cell energy imbalance and the activation of AMPK after Bt infection, we knocked down the *aak-2* transcription levels by RNAi in the mitochondria reporter strain SJ4143. The non-RNAi and *aak-2* RNAi worms were then fed with BMB171/Cry5Ba or BMB171/pHT304, respectively. The *aak-2* RNAi worms showed a high level of MF under BMB171/Cry5Ba treatment, but not under BMB171/pHT304 treatment (Fig. 4e, f). Next, we measured the AMP and ATP concentration

of wild-type *C. elegans* N2 and *aak-2* RNAi worms fed with BMB171/Cry5Ba or BMB171/pHT304. We found the AMP / ATP ratio was significantly increased in both N2 and *aak-2* RNAi worms fed with BMB171/Cry5Ba compared to BMB171/pHT304 (Fig. 4g). Our results further demonstrate that AMPK activation is a result rather than a cause of the energy imbalance. Thus, we concluded that Bt-mediated cell energy imbalance activates the AMPK in *C. elegans*.

**AMPK activation is involved in *C. elegans* defense responses against Bt infection**. Several previous studies reported that AMPK defends against low glucose levels, dietary deprivation, paraquat, physical stress, and pathogens[50–52]. Therefore, we wondered if AMPK activation was involved in defense against Bt infection. First, we compared previously identified AAK-2 target genes (total 1250 genes)[53–55] to our transcriptome analysis of gene expression in response to Bt infection, we found 305 genes up-regulated by Bt infection are also targets of AAK-2 (Fisher's exact test, *P* < 0.0001). The Phenotype Enrichment Analysis[56,57] according to the Wormbase database revealed several genes related to defense against pathogens (Fig. 5a and Supplementary Data 4). AMPK complexes consist of three subunits, including a catalytic subunit (α), and two regulatory subunits (β and γ). To confirm our hypothesis, we tested the sensitivity of four AMPK null alleles mutants and the wild-type N2 worms exposed to BMB171/Cry5Ba, including *aak-1(tm1944)* (subunit α1 of AMPK), *aak-2(ok524)* (subunit α2 of AMPK), *aakb-1(tm2658)* (subunit β1 of AMPK), and *aakg-4(tm5269)* (subunit γ1 of AMPK). Compared to the wild-type N2 worms, only *aak-2(ok524)* mutant worms showed increased sensitivity to BMB171/Cry5Ba infection (Fig. 5b, d). Survival assays confirmed that the *aak-2(ok524)* mutant worms are more sensitive to BMB171/Cry5Ba infection (Fig. 5c).

To further confirm the importance of AMPK catalysis subunit α2 (AAK-2 protein) in defense against Bt infection, we tested the sensitivity of null allele of *aak-2*, *aak-2(gt33)* worms, fed with BMB171/Cry5Ba. We found that *aak-2(gt33)* is more sensitive to BMB171/Cry5Ba than wild-type N2 (Supplementary Fig. 11). A similar phenotype was also observed in *aak-2* RNAi worms (Supplementary Fig. 12).

Next, we wondered whether the role of AMPK subunit α2 was unique to Bt infection. The sensitivity of the wild-type N2 and *aak-2(ok524)* mutant worms were analyzed to the heavy metal copper sulfate and oxidative stress (hydrogen peroxide). Our results showed that it is no significant difference between the wild-type N2 and the mutant *aak-2(ok524)* worms under these treatments (Supplementary Fig. 13).

Furthermore, we activated AMPK using AICAR which can activate AMPK via phosphorylation of AAK-2 Thr172 (Fig. 4a).

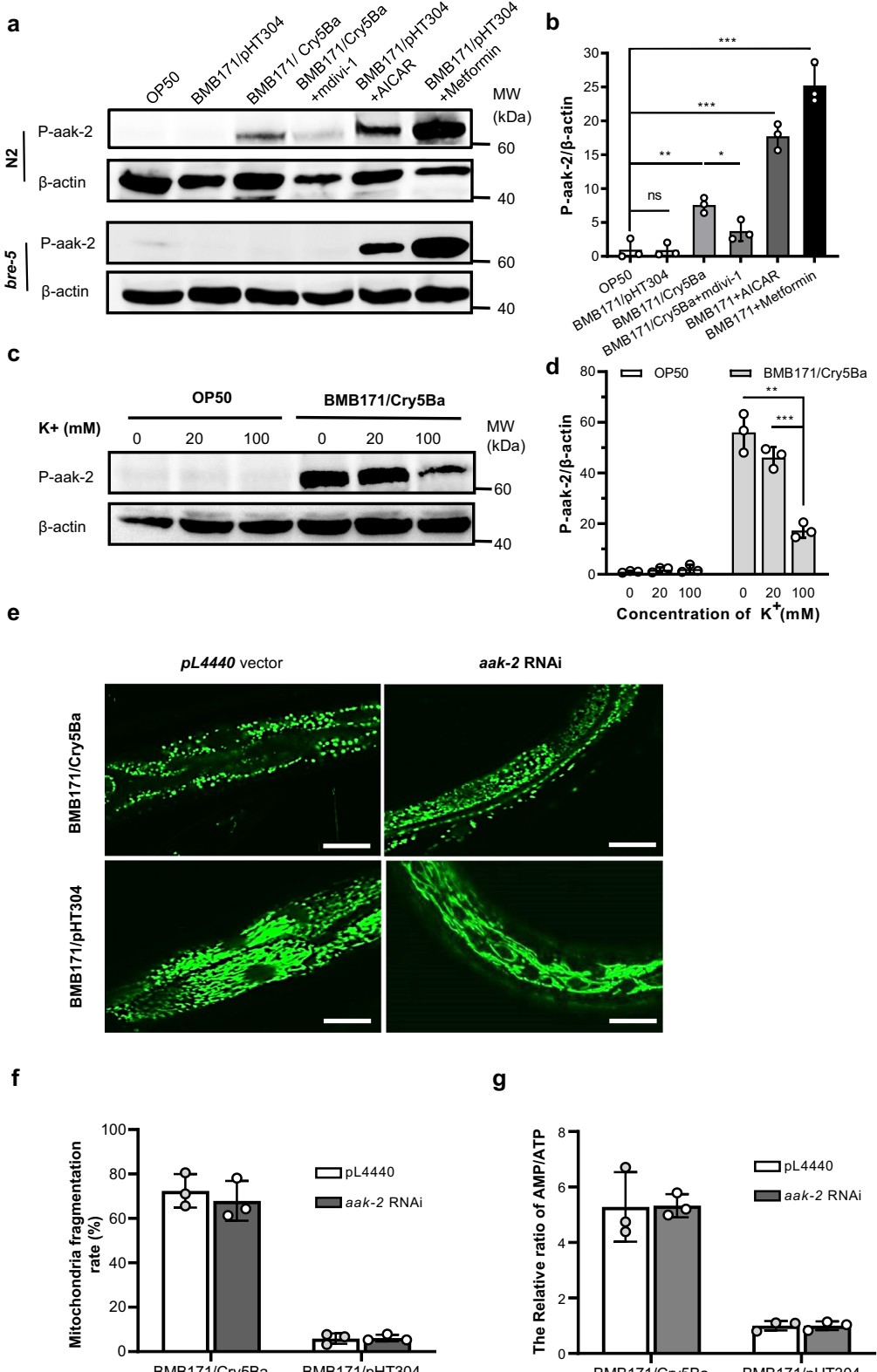

Then we compared the sensitivity of the AICAR-treated with non-treated N2 worms during BMB171/Cry5Ba infection by survival assays and growth assay. The results showed that the AICAR-activating worms showed more significant resistance to BMB171/Cry5Ba (Fig. 5e). However, compared with the wild-type N2 worms, the resistance to BMB171/Cry5Ba infection was not significantly different for *aak-2* mutant either in AICAR-activating or not (Fig. 5e, f). Above, we found that low concentrations of environmental potassium ions can enhance the resistance of worms to Bt, while high concentrations have the opposite effect (Fig. 3g), which corresponded to the level of AMPK activation. Taking together, our results demonstrated that AMPK plays an important role in *C. elegans* defend against BMB171/Cry5Ba.

**Fig. 4 AMPK was activated by nematicidal Bt infection. a** Western blotting showing the phosphorylation of the protein AAK-2 when exposed to BMB171/Cry5Ba after 4 h. Representative worms are shown. $N = 3$ independent experiments. **b** p-AAK-2 was quantitated by densitometry of protein bands from three independent experiments. β-actin was used as a loading control. $N = 3$ independent experiments. **c** Western blotting shows the phosphorylation of the protein AAK-2 when exposed to BMB171/Cry5Ba at different potassium concentrations. Representative worms are shown. $N = 3$ independent experiments. **d** Quantitative results of the phosphorylation level of AAK-2 relative to the concentration of β-actin. $N = 3$ independent experiments. The uncropped western blot images are shown as Supplementary Fig. 9. **e** Mitochondrial morphologies of transgenic *C. elegans* SJ4143 fed with BMB171/Cry5Ba or BMB171/pHT304. Representative images of worms treated by each strain are shown. $N = 3$ independent experiments. Scan bars represent 20 μm. **f** Analysis of the percentage of *aak-2* RNAi worms showing MF phenotype after treatment with BMB171/PHT304 and BMB171/Cry5Ba for 4 h. $N = 3$ independent experiments containing at least 50 worms each. **g** Analysis of the AMP/ATP ratio after treated by BMB171/pHT304 and BMB171/Cry5Ba without and with *aak-2* RNAi. $N = 3$ independent experiments. AICAR: a typical activator stimulating AAK-2 kinase activity; metformin: a compound to increase intracellular AMP; pL4440: empty vector RNAi control. Data points represent the mean values of three independent replicates, error bars denote the SD in (**b**), (**d**), (**f**), and **g**). The *p*-value was determined by a Student's *t*-test (Welch's correction for unequal variances). Error bars denote the SD, ****p* < 0.001, ***p* < 0.01, **p* < 0.05.

**AMPK activity in the intestine is required for *C. elegans* resistance to Bt infection.** To independently confirm the importance of AAK-2 in defense to BMB171/Cry5Ba infection, we constructed the transgenic strain *aak-2(ok524)* (P$_{aak-2}$::*aak-2*), which expresses *aak-2* under its native promoter P$_{aak-2}$ to rescue the *aak-2* function in *aak-2(ok524)* worms. The growth assay and survival assay results showed that *aak-2* expression driven by its own promoter P$_{aak-2}$ can completely alleviate the hypersensitivity of *aak-2(ok524)* mutant to BMB171/Cry5Ba infection (Fig. 6a, b), supporting that AAK-2 is independently important for worms defense against BMB171/Cry5Ba infection.

*C. elegans* lacks professional immune cells, yet can rely on epithelial cells for immune defenses[58]. Cry5Ba can attack the intestine of worms and form pores in the membrane of the intestine cell[59,60]. As a result, we hypothesized that intestinal-specific activity of AMPK regulates immune responses to Bt infection. We drove the *aak-2* expression under different tissue-specific promoters, including the intestine-specific promoter P$_{vha-6}$[61], the muscle-specific promoter P$_{myo-3}$[62], and the neuron-specific promoter P$_{rab-3}$[63]. We found that only *aak-2* expression under the intestine-specific promoter P$_{vha-6}$ alleviated the hypersensitivity of *aak-2(ok524)* mutant under BMB171/Cry5Ba infection (Fig. 6c, d). In contrast, there were no significant differences among the *aak-2(ok524)* mutant and the muscle-specific P$_{myo-3}$ or neuron-specific P$_{rab-3}$ rescued worms (Fig. 6c, d). In addition, we knocked down the *aak-2* gene by RNAi individually in the intestine, muscle, and epidermis of worms. Only the intestine-specific *aak-2* RNAi worms were more sensitive to BMB171/Cry5Ba infection (Fig. 6e), but not the epidermal or muscular-specific *aak-2* RNAi worms (Supplementary Fig. 14). These results demonstrate that the intestine serves as the first line of AMPK-mediated defense against BMB171/Cry5Ba attack.

**AMPK activation triggers DAF-16 and p38-MAPK-dependent innate immune signaling pathway during Bt infection.** The AMPK pathway is evolutionarily conserved from *C. elegans* to mammals, and it regulates many downstream pathways[64]. Here, we have shown that AMPK plays an important part in *C. elegans* defense against Bt infection. However, it was unclear whether known AMPK downstream genes or pathways are involved in *C. elegans* defense against Bt infection. AAK-2 has been reported to modulate the phosphorylation of the FOXO family transcription factor DAF-16[65]. DAF-16 can regulate many genes involved in metabolism, immune responses against several pathogens, and longevity of *C. elegans*[66,67]. Moreover, the DAF-16 was also triggered by nematicidal Bt infection[68], and functioned as an important modulator in defense against nematicidal PFTs in *C. elegans*[69]. Therefore, we speculated that AMPK may regulate the DAF-16-dependent signaling pathway in defense against Bt infection. To confirm this hypothesis, we compared previously identified DAF-16 target genes (class 1 genes)[66] with our RNA-

Seq data. We found 60 genes up-regulated by Bt infection are also the targets of DAF-16 (Fisher's exact test, $P < 0.0001$) (Fig. 7a, Supplementary Data 3, 5). We selected 6 genes (*thn-2*[70], *lys -7*[71], *clec-166*[72,73], *tre-4*[74], F32A5.4[75], and *sod-3*[71,76,77]) reported to be involved in the defense response and the most up-regulated gene (*ttr-44*[78,79]) to determine their transcription by qPCR. The transcription of these 7 genes was significantly up-regulated after Bt infection in wild-type N2 worms. Moreover, RNAi of *daf-16* or *aak-2* significantly suppressed the expression of these genes induced by Bt infection (Fig. 7b). Our results support that AMPK modulates the expression of downstream effectors in the DAF-16 pathway.

We further study the role of DAF-16 activation. DAF-16 is distributed predominantly throughout the cytoplasm of all tissues under standard growth conditions. DAF-16 will be phosphorylated and translocated from cytoplasmic to the nucleus when activated, where it binds to the promoter region and activates the expression of target genes[66]. We monitored the cellular translocation of DAF-16 using transgenic worms TJ356(I*sdaf-16*:: *gfp*) as a reporter, which expresses a functional DAF-16:: GFP fusion protein. Our results showed after BMB171/Cry5Ba infection, most of the DAF-16 was translocated from the cytoplasm to the nucleus in the intestine, especially the front and middle parts of the intestines (Fig. 7c, d and Supplementary Fig. 15). In contrast, the control strain BMB171/pHT304 failed to cause DAF-16 transfer to the nucleus at the same conditions (Fig. 7c, d and Supplementary Fig. 15). To test whether the AMPK activity is required for the activation of DAF-16 during Bt infection, we tracked the cellular translocation of DAF-16 when the *aak-2* gene was silenced by RNAi in TJ356. The observations showed that the DAF-16 nuclear translocation induced by Bt was significantly diminished by RNAi *aak-2* gene (Fig. 7c, d). Conversely, when we used mdivi-1 to inhibit the MF caused by Bt, the nuclear localization of DAF-16 was significantly restored. The metformin that activates AMPK by increasing intracellular AMP/ATP also significantly changed the localization of DAF-16 in worms fed by OP50 (Supplementary Fig. 15). Our results support that AMPK can trigger the DAF-16-dependent immune signaling pathway during Bt infection.

To test whether the immune-related genes we found are functioning in *C. elegans* defend against BMB171/Cry5Ba infection, we knocked down these seven genes by RNAi and then tested the sensitivity of these worms under BMB171/Cry5Ba treatment. Our results showed that RNAi of the genes except for *ttr-44* and *clec-166* lead to increased sensitivity to BMB171/Cry5Ba to varying degrees (Fig. 7e). This also strongly suggests that *C. elegans* can regulate these effectors through DAF-16 for defense against Bt infection.

To directly observe whether a downstream effector of DAF-16 responds to Cry5Ba infection, we tested the expression of SOD-3,

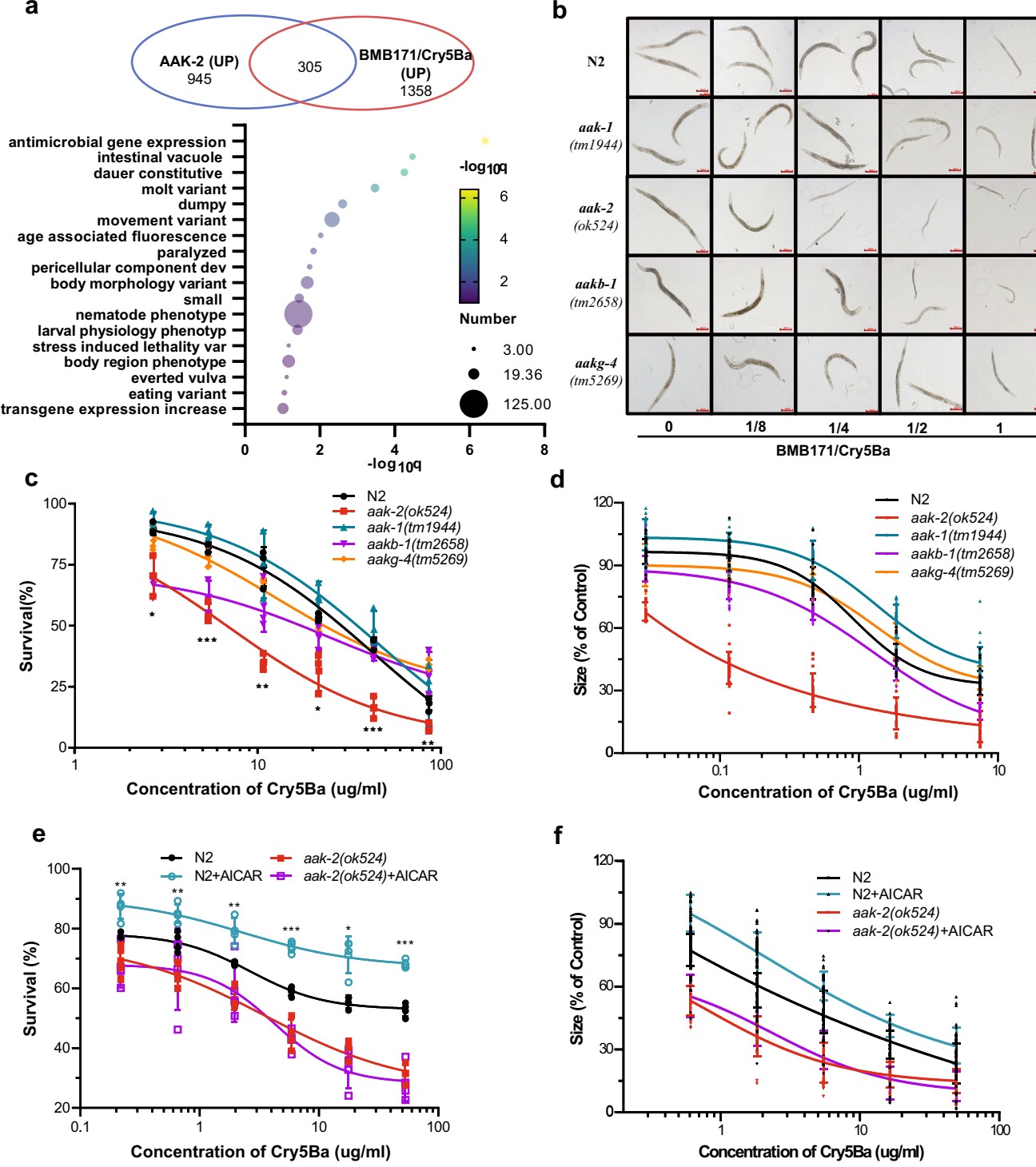

**Fig. 5 The AMPK activation is involved in *C. elegans* defense against Bt infection. a** Venn diagram comparing the overlaps between genes activated by nematicidal Bt and the target genes of AAK-2 in *C. elegans*. The Phenotype Enrichment Analysis according to Wormbase database was used to identify gene function. **b** The growth assay of the wild-type N2 and mutant worms after BMB171/Cry5Ba infection. The "0, 1/8, ¼, ½, and 1" represent the different concentrations of BMB171/Cry5Ba, up to 7.4 µg/ml Cry5Ba, as shown in (**d**). Representative worms are shown for each dose. Scale bar represents 200 µm. $N = 3$ independent experiments. **c** Survival of wild-type N2 and mutant worms after BMB171/Cry5Ba infection. **d** Quantification of worm growth. To quantify growth, worms were photographed under ×100 magnification for each dose, and the average worm areas were calculated. The size of the worms in the absence of toxin was set at 100%. The survival assay (**e**) and the growth assay (**f**) of the wild-type worms N2 and mutant worms *aak-2(ok524)* after BMB171/Cry5Ba infection when the AMPK was activated by AICAR or not. $N = 3$ independent experiments contain three replication of at least 30 worms in (**c**) and (**e**), $N = 3$ independent experiments containing at least 30 worms in (**d**) and (**f**). Data points represent the mean values of three independent replicates, error bars denote the SD in (**c**-**f**). The *p*-value was determined by a Student's *t*-test (Welch's correction for unequal variances), ***$p < 0.001$, **$p < 0.01$, *$p < 0.05$ and ns indicate no significant difference.

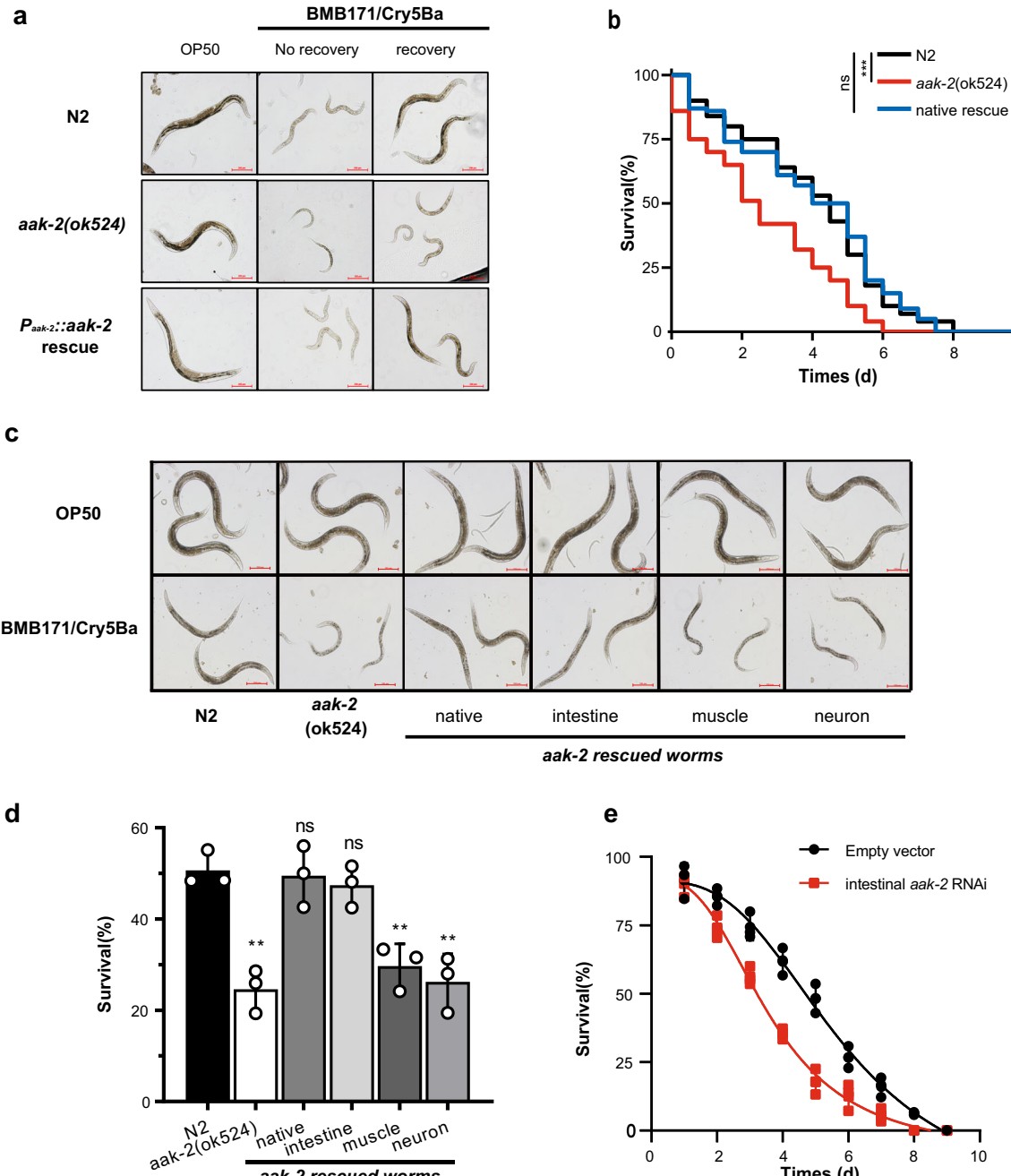

**Fig. 6 AMPK activity in the intestine is required for *C. elegans* resistance to Bt infection. a** Growth assay of *C. elegans* N2, *aak-2(ok524),* and P*aak-2*::*aak-2* rescued worms after fed with OP50 or BMB171/Cry5Ba for 30 min and then recovered on an *E. coli* OP50 NGM plate for 24 h or not. Representative images of worms in each condition are shown. Scale bar represents 200 μm. *N* = 3 independent experiments. **b** The survival assay of the *C. elegans* N2, *aak-2(ok524),* and P*aak-2*::*aak-2* rescue worms after being treated by BMB171/Cry5Ba (*n* = 50). Data were Kaplan–Meier analyzed followed by log-rank test. **c** The growth assay of the *C. elegans* N2, *aak-2(ok524),* and several kinds of rescue worms after being fed with OP50 or BMB171/Cry5Ba for 3 days. Representative images of worms in each condition are shown. Scale bar represents 100 μm. *N* = 3 independent experiments. **d** The survival of wild-type N2, *aak-2(ok524)* and several kinds of rescued worms after BMB171/Cry5Ba infection for 96 h. *N* = 3 independent experiments. **e** The survival of N2 worms after BMB171/Cry5Ba infection without or with intestinal *aak-2* RNAi. At least 30 worms were used in each repeat. *N* = 3 independent experiments. Data points represent the mean values of three independent replicates, error bars denote the SD in (**d**) and (**e**). The *p*-value was determined by a Student's *t*-test (Welch's correction for unequal variances), \*\*\**p* < 0.001, \*\**p* < 0.01, \**p* < 0.05 and ns indicates no significant difference.

a well-known direct target of DAF-16 during Bt infection[77,80]. We observed the expression of SOD-3 using transgenic worm CF1553(*muIs84*(*sod-3*::GFP))[80]. When the SOD-3::GFP reporter worms were exposed to BMB171/Cry5Ba, the expression of SOD-3 was significantly up-regulated compared to BMB171/pHT304. Meanwhile, the induction was significantly inhibited when either

the *daf-16* or *aak-2* gene was knockdown in the strain CF1553 (Fig. 7f and g). Taken together, we concluded that the AMPK triggers the DAF-16-dependent innate immune pathway to defend against Bt infection.

We next determined whether AAK-2 responds to Bt infection solely through the DAF-16 pathway. We knocked down *daf-16* or

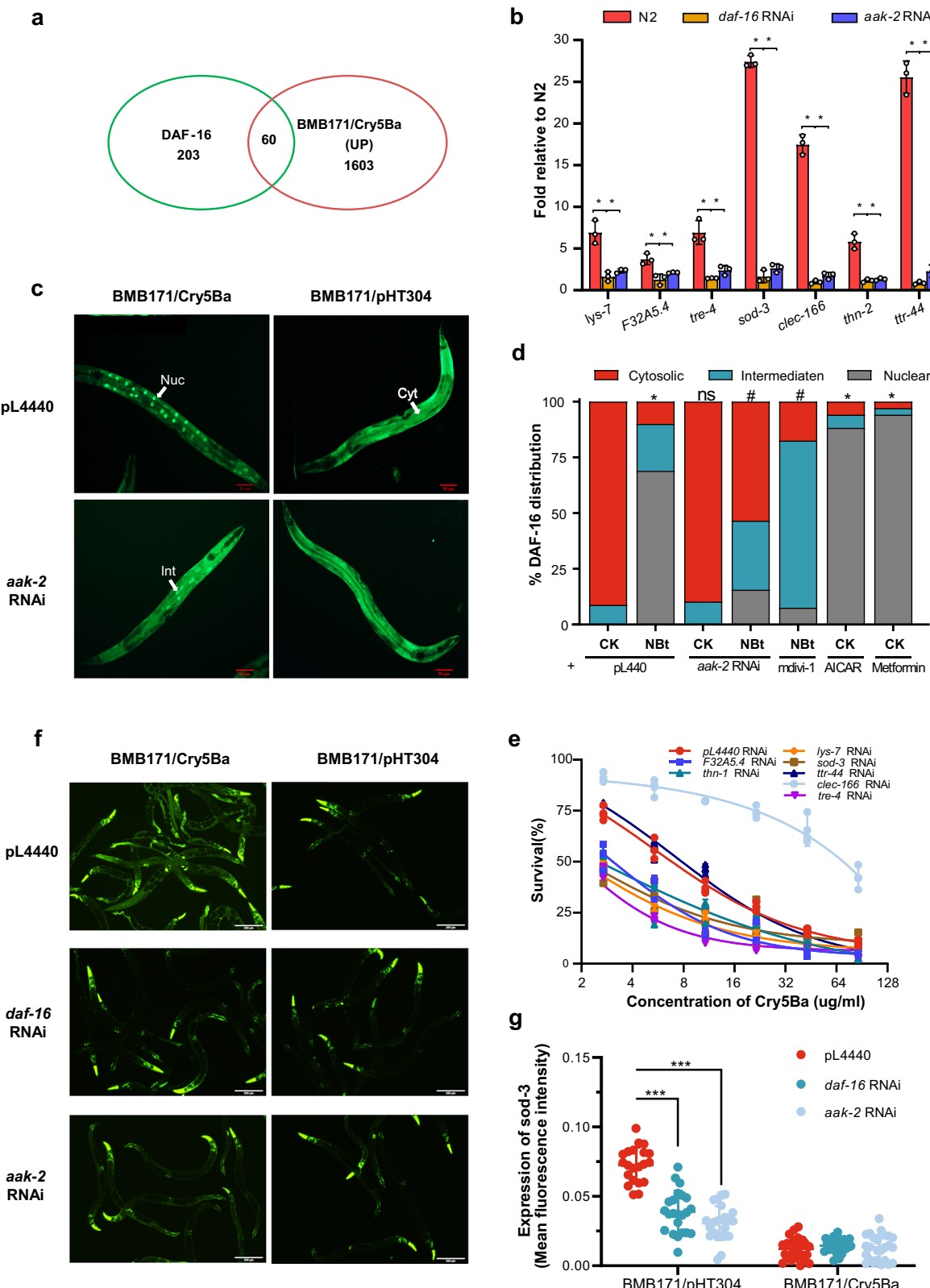

*aak-2* under *aak-2* or *daf-16* mutant worms. A survival assay showed the survival rate of the knockdown of *aak-2* under *daf-16* mutation was significantly lower than *daf-16* mutation individually (Supplementary Fig. 16a). Our results suggest that besides the DAF-16 pathway, other immune pathways may be involved in the defense against Bt. The MAPK pathway is also reported to be a downstream component of the AMPK signaling pathway in

mammal cells[81], which was reported to be an important immune pathway of *C. elegans* defense against Bt and PFTs, especially for Cry5Ba[59,82,83]. We also monitored the activation of mitogen-activated protein kinase (MAPK) pathways. We found the expression of 6 typical genes *pmk-1*, *tir-1*, *nsy-1*, *sek-1*, *kgb-1*, and *jun-1* which are involved in the p38-MAPK pathway were significantly up-regulated after Bt infection. Of these, three genes

**Fig. 7 AMPK triggers DAF-16-dependent innate immune signaling pathway to defense against Bt infection. a** Venn diagram showing the overlap in genes activated by nematicidal Bt and the target genes of DAF-16 in *C. elegans*. **b** qRT-PCR analysis of the expression of DAF-16 target genes in wild-type N2, *aak-2*(*ok524*), and *daf-16*(*mu861*) mutant worms after Bt infection. Error bars denote the Standard deviation. Data represent three biological replicates of representative results from three independent experiments with the same trend. **c** DAF-16 translocation observed during the no-RNAi and *aak-2* RNAi transgenic worms TJ356(*Isdaf-16::gfp*) fed with BMB171/Cry5Ba or BMB171/pHT304 for 2 h, respectively. The representative images for each treatment are shown. Scale bar represents 20 μm. $N = 3$ independent experiments. **d** The quantification of DAF-16 distribution after Bt infection (CK: BMB171/pHT304; NBt: BMB171/Cry5Ba). Worms with cytosolic (Cyt), intermediate (Int), or nuclear (Nuc) DAF-16 location were counted, and the percentages of each pattern of DAF-16 nuclear translocation were calculated. $N = 3$ independent experiments containing at least 30 worms. **e** The survival assay of the N2 worms after relevant gene knockdowns by RNAi under BMB171/Cry5Ba treatment for 4 days. $N = 3$ independent experiments contain three replication of at least 30 worms. Data points represent the mean values of three independent replicates, error bars denote the SD. **f** The expression level of P*sod*::GFP was observed using a reporter strain CF1553 (P*sod*::GFP) during the *aak-2* RNAi, *daf-16* RNAi, or no-RNAi after Bt infection. Scale bar represents 20 μm. Representative images of worms in each condition are shown. $N = 3$ independent experiments. **g** Quantification of P*sod*::GFP expression levels calculated by the fluorescence intensity of individual worms. Worms were treated for 4 h and then photographed. $N = 3$ independent experiments containing at least 20 worms. Each point represents a worm. The *p*-value was determined by a Student's *t*-test (Welch's correction for unequal variances) in (**b**), (**e**), and (**g**); and by a two-sided Fisher exact test in (**a**) and (**d**) (*: vs CK; #: vs NBt). ***$p < 0.001$, **$p < 0.01$, *$p < 0.05$ and ns indicates no significant difference.

were significantly suppressed after *aak-2* RNAi, demonstrating AAK-2 could regulate the p38-MAPK pathway (Supplementary Fig. 16b). Furthermore, using PMK-1::GFP reporter worms (PRJ112 [*pmk-1*::GFP]), we found that the expression of PMK-1 was significantly induced by Bt infection, and it was indeed inhibited by *aak-2* RNAi (Supplementary Fig. 16c). Our work suggests that Bt infection can also trigger the p38-MAPK immune pathway via AMPK.

## Discussion

Pathogen recognition and activation of the host innate immune system is critical to understanding pathogen-host interaction[84]. It is generally accepted that microbial infection can be recognized by host PTI. However, due to the widespread existence of PRR ligands, these conserve innate immune pathways are not sufficient to distinguish pathogens from certain probiotics or symbiotic bacteria. The ETI hypothesis is an important supplement to PTI and proposes that the host responds to pathogens by monitoring bacterial virulence factors[9,85]. Moreover, cellular surveillance systems, including the ribosome, proteasome, and mitochondria, monitor core cellular physiology activities serving as a novel pattern for hosts to distinguish pathogens from other microorganisms[19] and is considered to be an important part of the ETI response. However, how host cells use these systems to sense pathogens are unclear. There are several ways for the host to recognize the pathogens by monitoring core cell processes, including the DNA damage[15], the inhibition of translation[16], and the UPR^mt during the mitochondrial damage[86]. Here, we revealed that the leakage of potassium caused by pathogens leads to mitochondrial stress, which leads to severe energy imbalance, which is monitored by host cells to trigger innate immune responses via AMPK, secreting multiple effectors to defend against pathogens. Therefore, our work provides insights for the host cell to detect pathogens through cellular surveillance systems.

Previous works have shown several pathogens could attack and disrupt host mitochondria[17,87,88]. In our study, damage to the mitochondria was also detected when worms were infected by nematicidal Bt, including MF, a decrease in $\Delta\Psi_m$, and the changes in mtDNA content (Fig. 2). The damages to the mitochondria during Bt infection were mainly caused by its key virulent factor, Cry5Ba toxin (Fig. 2 and Supplementary Fig. 2). Although Cry5Ba could be translocated into epithelial cells, we did not find evidence of colocalization with mitochondria (Fig. 3a), indicating Cry5Ba may lead to mitochondrial dysfunction indirectly. Next, we confirmed that the leakage of intracellular potassium caused by Cry5Ba led to the mitochondrial damage directly (Fig. 3b).

This model is different from extensively studied toxins produced by *P. aeruginosa*[17] or *L. monocytogenes*[87].

Cellular energy imbalance is an important sign of mitochondrial disorder. In fact, several PFTs such as *Streptococcus* streptolysin O, *Vibrio cholera* cytolysin, *Staphylococcus aureus* α-toxin, and *Escherichia coli* hemolysin A, could also cause a decrease in intracellular ATP levels in a non-virally transformed human keratinocyte cell line[89]. Our results showed that Bt infection could cause a Cry-dependent cellular energy imbalance, which was widespread among nematicidal Bt Cry toxins (Fig. 1b and Supplementary Fig. 1). Both mitochondrial dysfunction and cytosolic energy imbalances were restored in a Cry5Ba-receptor mutant (*bre-5*), indicating toxin from nematicidal Bt is critical for these phenomena. Conversely, the mitochondrial division inhibitor mdivi-1, which can inhibit mitochondrial fragmentation but not the function of PFTs, can protect cells from AMP/ATP ratio imbalances (Fig. 2). In addition, the recovery of intracellular potassium concentration caused by Cry5Ba can also alleviate energy imbalances (Fig. 3c, d). Thus, the pores caused by the toxin are the primal reason for the changes in intracellular energy via potassium leakage. More importantly, this effect may be widespread in hosts infected by pathogens containing PFTs.

Previous research indicated that cytosolic energy imbalance and calcium content are important AMPK triggering patterns[90]. We found BMB171/Cry5Ba infection did not induce the change of cytoplasm calcium level of worms (Supplementary Fig. 6). However, the intracellular potassium leakage caused by nematicidal Bt led to mitochondria damage, which then gave rise to the dramatic increase of AMP/ATP ratio and AMPK activation via phosphorylation of α-catalysis subunit protein AAK-2 (Fig. 2a). Next, we found that the inhibition of mitochondrial fragmentation led to the recovery of the mtDNA content and the abnormal AMP/ATP content and also inhibited the activation of AMPK (Fig. 2). These data suggest that MF was the reason why the AMP/ATP content changed when the mitochondria had been damaged. We also found that Bt infection led to a decrease in mitochondrial membrane potential, which is generally known to be a consequence of mitochondrial ATP[91]. Our work also revealed that restoring potassium can not only restore mitochondrial damage, but also restore energy imbalance and AMPK activation. Together, our work revealed Bt infection increased the rate of AMP/ATP and cytosolic energy imbalance, which ultimately triggers AMPK via AAK-2 phosphorylation.

Accumulating evidence suggests that AMPK is not only a crucial evolutionarily conserved cellular energy sensor but also plays an important role in host-pathogen interactions[51]. Our work proved the activation of AMPK is helpful for *C. elegans* to defend against nematicidal Bt (Fig. 5). What's more, AMPK

activity in intestinal cells is particularly important for *C. elegans* against nematicidal Bt (Fig. 6). As we know, due to the lack of professional immune cells, the *C. elegans* intestine is considered to be the first line to identify and defense against pathogens[92]. Therefore, we can infer that the host cell monitors the energy imbalance caused by pathogens through mitochondria, and then activate a series of responses against pathogens. Our work stressed the importance of the connection between the cellular mitochondria surveillance systems and AMPK through energy changes during pathogen infection.

In addition, how mitochondria disorder triggers downstream innate immunity signaling is largely unknown yet. Accumulating evidence indicates AMPK regulates DAF-16 and MAPK-dependent pathway which is related to defense response and functions to extend lifespan[50,65,81,93,94]. Here, we found that AMPK activation mediated by mitochondrial damage regulates the DAF-16 to increase the transcription of certain DAF-16-dependent innate immune effectors (Fig. 7). The different sensitivity of worms against Bt infection indicates that the DAF-16 is not the only immune pathway activated by AMPK (Supplementary Fig. 16a). Our results proved AMPK also triggered the p38-MAPK innate immune pathways apart from DAF-16 during Bt infection (Supplementary Fig. 16b, c). We have observed that *C. elegans* can produce a variety of antibacterial substances through the DAF-16 pathway to defend against BMB171/Cry5Ba infection, such as lysozyme and thaumatin. In addition, a parasite aspartyl protease inhibitor and several effectors related to environmental resistance such as superoxide dismutase (SOD), and trehalase also help worms defend against Bt infection (Fig. 7e). Interestingly, when we knockdown the *clec-166* which encodes C-type lectin, *C. elegans* showed very strong resistance against Bt. However, Pees et al. also reported the opposite effect of C-type lectin in *C. elegans* response to pathogens[95]. The specific function of C-type lectin in *C. elegans* deserves further study. Altogether, our results demonstrate that *C. elegans* can activate multiple immune responses to defense against Bt infection.

In conclusion, our work demonstrated that host cells can directly sense mitochondrial-mediated intracellular energy imbalance to monitor pathogens, activating downstream innate immune responses to defense pathogens through AMPK. As modeled in Fig. 8, nematicidal Bt infection in *C. elegans* caused potassium leakage, leading to the mitochondria damage and the energy imbalance of intestinal epithelial cells, the latter of which subsequently triggered AMPK *via* phosphorylation of AAK-2. Then, the AMPK modulates DAF-16-dependent and p38-MAPK-dependent innate immune pathways to defend against nematicidal Bt infection. These findings revealed AAK-2 could work as a sensor and regulator of the *C. elegans* response to pathogens. Considering that PFTs produced by pathogens can generally lead to potassium ion leakage, as well as the mitochondria surveillance systems and AMPK are conserved components from worms to mammals, our study suggests that disrupting mitochondrial homeostasis to activate the immune system through AMPK-dependent pathways may widely exist in animals, which may provide extended strategies for immunotherapy of multiple diseases.

## Methods

**Caenorhabditis elegans and bacterial strains**. *Caenorhabditis elegans* strains used in this work were kindly provided by the *Caenorhabditis* Genetics Center (CGC) or the National Bioresource Project (NBRP) and listed in Supplementary Table 1. All *C. elegans* strains were maintained on nematode growth media (NGM, 0.3% NaCl, 0.25% tryptone, and 1.5% agar) using *E. coli* OP50 as food under standard conditions[96]. The bacterial strains and plasmids used in this study are also listed in Supplementary Table 1. All *Escherichia coli* and Bt strains were grown on Luria-Bertani (LB) medium supplemented with the appropriate antibiotics at 37 °C or 28 °C, for *E. coli* or Bt, respectively. BMB171/Cry5Ba, BMB171/Cry5Ca,

BMB171/Cry21Aa, BMB171/Cry6Aa, and BMB171/Cry1Ac used in this work are derived from the acrystalliferous Bt mutant BMB171 transformed with toxin gene *cry5Ba*, *cry21Aa*, *cry6Aa*, and *cry1Ac* on shuttle vector pHT304[97,98], respectively. BMB171/pHT304 used in this work is the acrystalliferous Bt mutant BMB171 transformed with an empty shuttle vector pHT304.

**Construction of gene aak-2 rescued worms**. A 2.1 kb fragment of the intestine-specific *vha-6* promoter ($P_{vha-6}$), a 2.5 kb fragment of the muscle-specific *myo-3* promoter ($P_{myo-3}$), a 1.3 kb fragment of the neuron-specific *rab-3* promoter fragment ($P_{rab-3}$) and a 3.0 kb fragment of *aak-2* promoter fragment ($P_{aak-2}$) were generated by PCR from total DNA of *C. elegans*, then inserted into pPD49.26 to construct 4 recombinant plasmids of pPD49.26-P. A full-length 1.8 kb of the *aak-2* cDNA and a 1.6 kb of the 3'-UTR of *aak-2* were generated by overlap PCR and inserted into pPD49.26-P at the downstream of the promoters, respectively. The recombinant plasmids which contained gene *aak-2* with the tissue-specific promoters, $P_{aak-2}$::*aak-2*, $P_{vha-6}$::*aak-2*, $P_{myo-3}$::*aak-2*, and $P_{rab-3}$::*aak-2*, were injected into the gonads of *aak-2(ok524)* worms to generate independent transgenic lines by standard germline transformation techniques[99]. All of the 4 fusion genes were injected at concentrations ranging 100 ng/μl with $P_{lin-44}$::GFP at 100 ng/μl as a co-injection marker.

**RNA interference**. RNAi *E. coli* strains containing targeting genes were originally delivered from the Ahringer RNAi library[100] and were kindly provided by Professor Zhenxing Wu, Huazhong University of Science and Technology. RNAi feeding experiments were performed as described before[100] on synchronized L1 larvae at 20 °C for 40 h.

**Quantitative real-time RT-PCR analysis**. Total RNA isolated using Trizol reagent (Invitrogen) was reversely transcribed with random primers using Superscript II reverse transcriptase (Invitrogen) according to the manufacturer's protocol. Real-time PCR analysis was performed with Life Technologies ViiA™ 7 Real-Time PCR system (Life Technologies, USA) using Power SYBR Green PCR Master Mix (Life Technologies, USA). The gene *tba-1* was used as an internal reference[101]. The primers used for PCR are listed in Supplementary Table 2. The experiments were conducted in triplicate, and the results were expressed as $2^{-\triangle\triangle Ct}$.

**Nucleotide measurements**. The AMP, ADP, and ATP were extracted by the adapting perchloric acid method[102]. 300–400 worms were washed with ice-cold M9 buffer, resuspended in 20 μl of M9 buffer, to which was added 40 μL of ice-cold 8% (v/v) $HClO_4$ on ice. The worms were then homogenized in liquid nitrogen, and the homogenates were disrupted using ultrasonic vibrations and neutralized with 1 N KOH. Next, the suspension was centrifuged (10,000 × *g* for 3 min at 4 °C), and the supernatant was passed through a 0.2 μm filter (Nanosep). The concentration of AMP, ADP, and ATP was detected using the LC-MS (Agilent Technologies, 1260-6540) based on the published method[103,104].

**Analyses of mitochondrial morphology**. SJ4143(*zcIs17*[$P_{ges-1}$::GFP^mt]) worms stably express GFP in mitochondria matrix of intestinal cells and can be used as a reporter for mitochondrial morphology. The worms were synchronized according to the method described before[40]. We define the mitochondria fragmentation (MF) phenotype when more than 50% of the mitochondria in the worm are punctate. Then the worms were treated according to the liquid assay noted. Washed worms were collected and photographed by Olympus BX63 with excitation at 488 nm and emission at 525 nm. At least three repeats were performed for each condition with at least 50 animals photographed per treatment.

**Measurement mitochondrial membrane potential (ΔΨm)**. Tetramethyl rhodamine ethyl ester (TMRE) ((MCE, # 115532-52-0)) is a lipophilic cation to detect the membrane potential (ΔΨm) of the mitochondrial[105]. The uncoupling agent carbonyl cyanide 4-(trifluoromethoxy)phenylhydrazone (FCCP) a protonophore disrupt ATP synthesis due to the collapse of H+ gradient, resulting in a decrease in mitochondrial membrane potential Synchronized N2 worms were treated according to the liquid assay. After being treated by respective bacteria and compounds, worms were resuspended in M9 containing 8 μM TMRE. After 2 h treatment, worms were washed with M9 for 3 times to remove dye thoroughly. Worms were photographed using Olympus BX63 epifluorescence microscope at ×10 magnification with excitation at 550 nm and emission at 575 nm. To quantify the mitochondrial membrane potential, we converted the fluorescence intensity of each worm to the optical density value and used the cumulative optical density/ worm area as the quantitative value of membrane potential intensity. At least three repeats were performed for each condition and at least 20 animals were measured per treatment. The average optical density of each worm was measured using Image Pro Plus v6.0.

**Mitochondrial DNA (mtDNA) quantification assay**. MtDNA damage was assessed using real-time qPCR according to published methods[106,107]. The pair of primers (Forward: GTTTATGCTGCTGTAGCGTG, Reverse: CTGTTAAAG-CAAGTGGACGAG) corresponding to the mitochondrial genome were used to normalize to genomic DNA using primer pairs specific for *ama-1* (Forward:

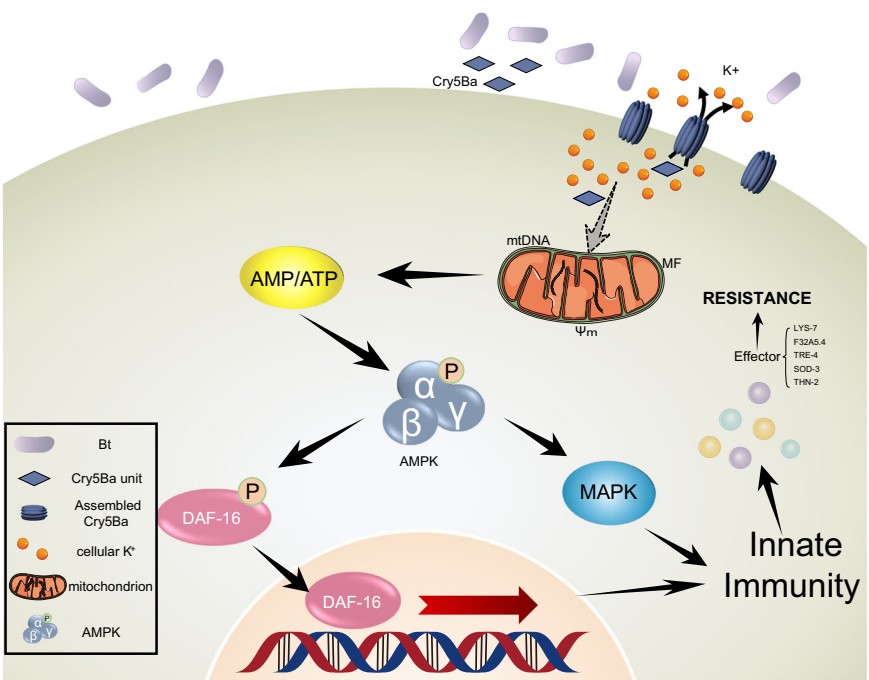

**Fig. 8 The action model of mitochondria surveillance systems for recognition of pathogens and the activation innate immune responses by AMPK in *C. elegans*.** After nematicidal Bt and crystals mixture are fed to the *C. elegans*, the Cry5Ba toxin is transported into the intestinal epithelial cell plasma membrane and assembled to trigger pore formation, resulting in the potassium ion leakage. The potassium ion leakage caused by Cry5Ba disrupts mitochondria, which indirectly leads to MF, a decrease in mitochondrial membrane potential, and the changes in mtDNA content, ultimately resulting in cellular energy imbalance. AMPK senses Bt-induced intercellular AMP/ATP changes and then phosphorylates AAK-2. The activated AAK-2 modulates DAF-16 and p38-MAPK-dependent innate immune pathways to defend against pathogens. Solid arrows represent relationships confirmed in this study. The dotted arrow represents an indirect relationship.

TGGAACTCTGGAGTCACACC, Reverse: CATCCTCCTTCATTGAACGG). Synchronized worms were treated and collected as described above. Collecting worms were homogenized in liquid nitrogen and lysed in a standard buffer containing proteinase K for 1 h at 65 °C. The RT-qPCR procedure was implemented as described above.

**Calcium and potassium imaging.** Fluo-4 AM (Molecular Probes) and ION Potassium Green-2 AM (APG-2 AM, Abcam) are used to detect the concentration of $Ca^{2+}$ and $K^+$, respectively, according to the published methods[108,109]. For $Ca^{2+}$ measurements, the treated *C. elegans* were loaded with 100 μM Fluo-4 AM for 1 h in M9 buffer. *C. elegans* were washed >3 times and then photographed by Olympus BX63 epifluorescence microscope. For $K^+$ measurements, treated *C. elegans* were labeled in 5 μM APG-2 AM in each buffer of different potassium concentrations for 1 h. The labeled *C. elegans* were cultured in the corresponding buffer with added OP50 for 1 h to remove the free dye in the intestine. The washed *C. elegans* were photographed under the same conditions. The excitation and emission wavelengths of Fluo-4 AM were 489 and 508 nm. For APG-2 AM, they are 526 and 550 nm, respectively. At least three repeats were performed for each condition and at least 20 animals were measured per treatment. The average optical density of each worm was measured using Image Pro Plus v6.0.

**Bt infection assay.** The Bt infection assays were based on the published protocol[110] which contains two different procedures. The plate assays: Bt strains were grown in LB medium containing 50 μg/ml spectinomycin at 28 °C overnight and seeded 400 μl to a fresh NGM plate. Then these plates were placed at 28 °C for 3 days. Synchronized nematodes were grown to the L4 stage in NGM plate containing OP50. Nematodes were then washed off and cleaned by M9 buffer. 2000–3000 worms were added to the Bt plate and treated at 25 °C for 4–6 h. The liquid assay: Synchronized L1 worms were grown to L4 stage on OP50 plate at 20 °C. 300–500 worms were then added to a 24-well plate (total volume 50 μl) containing 400 μl M9 buffer, to which we added the different mixture of crystal toxins and spores in each well (total volume 50 μl). The plates were placed at 25 °C for 4–6 h.

**Growth assays.** L1 growth assays were carried out using different doses of the mixture of crystal Cry5Ba and spores as described before[110], *E. coli* OP50 was added at an optical density 600 ($OD_{600}$) of 0.2–0.25 along with 20–30 synchronized

L1 larvae per well. After three days at 20 °C, at least 60 worms were photographed in different doses on a microscope, and then we calculated the average area for each condition using the software NIH ImageJ 1.33, normalizing the average area at each toxin concentration to the average area of the no toxin control. The size of the worms in the absence of toxin was set at 100%. Each experiment was independently replicated at least three times.

**Survival assays.** *C. elegans* N2 and mutant worms were exposed to the mixture of crystal Cry5Ba and spores in S media in 96-well plates to quantitatively score the survival as described before[110]. Concentrations of each toxin were set up in triplicate for each assay with ~20–30 synchronized L4 worms per well. Motility was used to determine whether worms were alive or not. Crawling worms were marked as alive. Non-crawling worms were gently touched with a platinum pick to observe their movement. The survival rate of each well was scored after incubating at 20 °C for 6 days. Each experiment was independently replicated at least three times.

**Lifespan assays.** Life span assays were performed at 20 °C on NGM plates. Each bacterium was fully spread on the NGM plate to prevent worms from avoiding or escaping the bacterial lawn. Approximately 60 L4-stage worms were incubated on NGM plate seeded with OP50 or other pathogens. Every plate contained 0.05 mg/ml of 5-fluorodeoxyuridine (FUDR) to prevent eggs from hatching. Five plates were tested for each strain in each experiment. Each experiment was independently replicated at least three times. The determination of whether worms were alive was as described for the survival assay of *C. elegans*. The surviving worms on each plate were counted at 20 °C every 12 h. Statistical analyses were assessed by Kaplan–Meier survival analysis followed by a log-rank test. Statistical significance was assessed by Kaplan–Meier survival analysis followed by a log-rank test.

**CuSO₄ and H₂O₂ assay.** $CuSO_4$ and $H_2O_2$ assays were conducted as previously described[111]. The survival assays were conducted in serial doses of $CuSO_4$ or $H_2O_2$ from 0.13 to 32 mM. 20–30 L4 synchronized worms were used per well in 48-well plates along with *E. coli* OP50 at an OD600 of 0.2–0.25. The survival rate in each well was determined after 6 days of $CuSO_4$ or 4 h of $H_2O_2$ exposure at 20 °C.

**Cry5Ba localization assay.** The L4-staged transgenic worms RT311[P$_{vha-6}$::GFP::RAB-11] (apical recycling endosome reporter worm[112]) and SJ4143(*zcIs17*

[P$_{ges\text{-}1}$::GFP$^{mt}$]) (mitochondria reporter worm that can stably express GFP in mitochondria matrix of intestinal cells[40]), were fed with rhodamine-labeled crystal protein Cry5Ba for 4 h. Then these worms were placed on 2% agarose pads and rhodamine-labeled and GFP-labeled signals were observed using the using confocal microscope at ×100 magnification. At least 2–3 independent biological repeats were carried out for each experiment.

**DAF-16 nuclear localization assay**. After 2 h of treatment with Bt, synchronized L4 TJ356 worms (transgenic animals expressing DAF-16::GFP) were immediately placed in M9 buffer and onto microscope slides. GFP localization was observed using a fluorescent microscope (Olympus BX31, Japan) at ×40 magnification. DAF-16 localization was categorized as cytosolic localization, intermediate localization, and nuclear localization[76]. The number of worms with each type of nuclear translocation was counted. Worms exposed to non-nematicidal Bt BMB171/pH304 were used as negative controls; while the worms exposed to heat shock for the same periods at 30 °C were used as a positive control. $P$ values were calculated using SPSS ver13.0 (SPSS, Chicago, IL).

**Western blotting analysis**. After feeding nematicidal Bt or non-nematicidal Bt, *C. elegans* N2 and different mutants were washed three times with ice-cold M9 buffer and then homogenized in liquid nitrogen. The homogenates were harvested in lysis buffer (50 mM HEPES, 150 mM NaCl, 10% glycerol, 1% Triton X-100, 1.5 mM MgCl$_2$, and 1 mM EGTA) with protein inhibitors (0.2 mM Na$_3$VO$_4$, 1 mM NaF). The lysate samples were subjected to SDS-PAGE using 10% (wt/vol) poly-acrylamide gels and proteins transferred to PVDF membranes (Life technologies). The trans-blotted membrane was washed three times with PBS containing 0.05% Tween 20 (PBST). After blocking with PBST containing 5% BSA, the membrane was probed with the diluted 1000 × primary antibody (Phospho-AMPKα(Thr172) Rabbit mAb, Cell Signaling, # 2535, or β-actin antibody, Proteintech, # 66009) and washed three times. The membrane was then probed with diluted 5000 × HRP-coupled secondary antibody (Proteintech, SA00001-2, # SA00001-1) and washed. Finally, the membranes were exposed using a chemiluminescent substrate (SuperSignal West Pico, Thermo Scientific).

**RNA-sequencing and transcriptome analysis**. Total RNA was isolated using Trizol reagent (Invitrogen, USA) according to the manufacturer's protocol. Total RNA concentrations and integrity were measured with NanoDrop 2000C spectrophotometer (Thermo Scientific, USA). The mRNA of each sample was purified using poly-dT oligo attached magnetic beads, and then fragmented (~200 bp) at an increased temperature. The first-strand cDNA was synthesized with random oligonucleotides and SuperScript II (Invitrogen, USA). The second-strand cDNA was synthesized by DNA polymerase I and RNase H. The second-strand cDNA was purified with Vahtstm DNA Clean Beads. Ends were repaired and 3-end single nucleotide A (adenine) was added. All RNA-seq libraries (non-strand-specific, paired-end) were prepared with the TruSeq RNA Sample Prep kit (Illumina, USA). Approximately 150 nt of the sequence was determined from both ends of each cDNA fragment using the HiSeq platform (Illumina) according to the manufacturer. After sequencing, clean reads were obtained by removing low-quality, adapter-polluted and high content of unknown base (N) reads. Then the clean reads were used to perform de-novo assembly with Trinity (https://github.com/trinityrnaseq/trinityrnaseq/wiki); TGICL (http://sourceforge.net/projects/tgicl/files/tgicl%20v2.1/) was used to further assemble all the unigenes from the two different groups to form a single set of non-redundant unigenes (called all-unigenes). Sequencing reads were annotated and aligned to the *C. elegans* reference genome using Tophat2. The alignment files from TopHat2 were used to generate read counts for each gene. GO enrichment analysis was implemented by mapping each differentially expressed gene into the records of the GO database (http://www.geneontology.org/). GO terms with corrected *p*-values <0.05 were considered to be significantly enriched by differentially expressed genes. Pathway enrichment analysis was based on the KEGG database according to KOBAS. (http://kobas.cbi.pku.edu.cn/)[113]. Biologically complex responses of genes and pathway annotation for unigenes were analyzed using KEGG annotation. A Q-value ≤0.05 was identified as significant enrichment of a pathway among the differentially expressed genes.

**Statistics and reproducibility**. All sample sizes were determined by previous studies. Sample sizes (number of worms) of fluorescence quantitative analysis are shown in the figure legend. Sample sizes $n = 3$ were used for transcriptome analysis, qRT-PCR, and survival assay[76]. For worms growth assay, at least 20 worms were used in each condition[110]. The western blot experiments are repeated three times. At least two biological replicates and three technical replicates are used for LC/MS data collection[102]. All data analysis was performed using SPSS, ver20.0 (SPSS, Chicago, IL, USA) or GraphPad Prism (version 8.0; GraphPad Software, La Jolla, California). Statistical analysis used either unpaired *t*-test or one-way ANOVA with Dunnett adjustment. Fisher's exact test was used to test for significant overlap between different gene sets. More details were shown in Supplementary Data 6. Statistics indicated are *$p < 0.05$, **$p < 0.01$, ***$p < 0.001$. The lack of any symbol indicates no significant difference.

**Reporting summary**. Further information on research design is available in the Nature Research Reporting Summary linked to this article.

## Data availability

Source data for this manuscript are provided as Supplementary Data 1. The uncropped western blot images are shown in Supplementary Fig. 9. The mRNA-sequencing data are available on the NCBI Sequence Read Archive (SRA) (https://www.ncbi.nlm.nih.gov/sra), under the bioproject PRJNA662857.

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

## Acknowledgements

This work was supported by the fund from the National Natural Science Foundation of China [U20A2040 and 31970076 to M.S., 31770116 to D.P., and 31600006 to S.J.]; the College Excellent Youth Science and Technology Innovation Team Project of Hubei Province [T201535 to S.J.]; and National Institutes of Health/National Institute of Allergy and Infectious Diseases grants R01AI056189 to R.V.A. We thank Professor Zhengxing Wu for providing RNAi strains (College of Life Science and Technology, Huazhong University of Science and Technology), as well as the Caenorhabditis Genetics Center for the worm strains. Microscopic image and some other data were acquired at the State Key Laboratory of Agricultural Microbiology Core Facility and National Demonstration Center for Experimental Biology Education (Huazhong Agricultural University).

## Author contributions

S.J. conceived the project, S.J. and H.C. designed and performed experiments. S.W., J.L., and Y.M. assisted to perform the experiments. S.J. and H.C. completed the writing of the manuscript. R.V.A. assisted to interpret the data and revise the paper. D.P. and M.S. designed and supervised the study, as well as revised the paper.

## Competing interests

The authors declare no competing interests.
