## [Peer Review File · Communications Biology]

Reviewers' comments:

Reviewer #1 (Remarks to the Author):

Ju et al. study *C. elegans* innate immune response to *B. thuringiensis*. The authors concluded that toxins from BT cause potassium leakage and imbalance of energy via mitochondrial damage. The authors also proposed AMPK-dependent surveillance system as a new strategy for host defense. The study is generally interesting and helpful. However, there are some analysis details missing that would help researchers understand the study.

Major comments:

- 1) L321 (Fig. 7A), author indicates p value is obtained from t-test. It seems not possible for application of t-test on overlap of two gene sets. As I know, common tests for such topic should be Fisher exact test or permutation test.
- 2) Author declared that worms triggered AMPK to defend against BT infection. It will be better if author could perform additional gene set enrichment analysis against AMPK pathways (such as *aak-2* study from <http://www.ncbi.nlm.nih.gov/pubmed/25723162>, <http://www.ncbi.nlm.nih.gov/pubmed/21331044> and <http://www.ncbi.nlm.nih.gov/pubmed/26959186>
- 3) L351 (Fig. 7B) author selected 8 genes from intersection between BT induced genes and *daf-16* targets and declared as "immune-related genes". However, there is no evidence supporting these genes are immune related (especially for *sod-3*).
- 4) Author confirmed that potassium can reverse the mitochondrial damage caused by BT toxin. Is it also involved in host defense against BT?

Minor:

- 1) Fig. 7A for 263 *daf-16* target genes. Author should mention that gene set is from class 1 genes in Murphy et al since Murphy et al provide two gene sets in their study.
- 2) Fig 6B. Author should indicate how many worms used in the survival assay.

Reviewer #2 (Remarks to the Author):

Brief summary of the manuscript

Ju et al. study the response of *C. elegans* to infection with the *Bacillus thuringiensis* (Bt) strain BMB171/Cry5Ba, which produces the pore forming toxin Cry5Ba. They demonstrate that infection leads to mitochondrial damage and an increased AMP/ATP ratio. Also, the authors show that Bt infection, and in particular exposure to the Cry5Ba toxin, leads to a decrease in potassium ion concentration. Since restoration of intracellular potassium reduced mitochondrial damage and restored the AMP/ATP ratio, the authors conclude that the leakage of potassium caused by exposure to the PFT Cry5Ba leads to mitochondrial damage and subsequently energy imbalance. Furthermore, the authors demonstrate that the energy sensor AMP-activated protein kinase (AMPK) is activated by BMB171/Cry5Ba infection downstream of mitochondrial damage and energy imbalance. The *aak-2(ok524)* mutation that affects the subunit $\alpha 2$ of AMPK, a *aak-2* null mutation, and RNAi silencing of *aak-2* reduced worm survival upon BMB171/Cry5Ba infection and the authors show that *aak-2* functions in the intestine to regulate resistance. Finally, the authors provide evidence of DAF-16-mediated transcriptional activation of gene expression downstream of *aak-2*.

Overall impression of the work

The findings presented in this manuscript are potentially significant and of broad interest. The authors identify *aak-2* as novel regulator of the *C. elegans* response to *B. thuringiensis* PFTs and link *aak-2* to DAF-16-dependent signaling. Moreover, they may have identified a novel potential mechanism of effector-triggered immunity (ETI) in *C. elegans*. Their results suggest that *aak-2* acts as 'guard

protein' that surveils the energy status of the cell to sense pathogen-mediated mitochondrial damage. The importance of effector-triggered immunity, which is mediated by the surveillance system and not by pattern recognition receptors, for animal defense responses to pathogen attack was established rather recently and our understanding on the role of guard proteins and the processes that they guard in metazoans is still emerging. This paper is thus very timely. However, to demonstrate that *aak-2* functions in ETI, the authors would need to show that *aak-2* is required to activate immune defenses against BMB171/Cry5Ba. They demonstrate that *aak-2* is required for increased survival after infection, but is this really the consequence of an activated pathogen defense response? What is unclear to me is what constitutes the defense response to BMB171/Cry5Ba infection, e.g. which genes are part of the defense response? The authors make first steps into that direction by investigating the expression of DAF-16 target genes (figure 7B) in the *aak-2* mutant background, but they do not explain the role of these genes in pathogen defense (or why they were chosen). They also investigate activation of the superoxide dismutase gene *sod-3*, but it remains unclear if it is required for defense against BMB171/Cry5Ba and the authors neglect to discuss its known function in defense against another *B. thuringiensis* strain (see comment #24 below). In line 426-427 of the discussion the authors state that "[...] transcription of certain DAF-16-dependent innate immune effectors, which helps *C. elegans* defend against pathogen infection." The claim of the immune effector function of the analyzed genes is not supported by the experiments done. It is thus unclear if *aak-2* really acts in ETI/surveillance immunity.

Another ETI mechanism in metazoans, including *C. elegans*, is surveillance of protein translation. Chemical inhibition of translation induces pathogen defenses in the absence of a pathogen. It would thus be very interesting to test if a decrease in AMP/ATP ratio in the absence of a pathogen induces pathogen defenses and this would also support the idea of *aak-2* mediated ETI. Did the authors consider this?

In addition, I have two major concerns:

1. Statistical analysis:

While the experiments were apparently done in a convincing and thorough manner (e.g. they authors used several *aak-2* mutant alleles, RNAi, and performed tissue-specific rescue experiments to demonstrate a role of *aak-2* in survival upon BMB171/Cry5Ba infection) and seem to be technically sound, I am concerned about the statistical analyses of the data. The only statistical tests the authors use to analyze their data are the t-test and one-way ANOVA, both parametric tests. In the case of survival analysis, both tests are not suitable (see comment #22 below) and with the same experimental design they use the t-test in one case and one-way ANOVA in another case (data shown in figure 5C and 6B). Did the authors check if their data are really all normally distributed? Also, the authors mainly present pooled data (i.e. the mean and SD of three independent replicates are shown). They should provide data and results of statistical tests (p-values from all experiments) for each experimental run in a supplementary table.

2. Writing:

The manuscript is not clearly written. It contains many grammatical errors and is difficult to read. The manuscript requires revision for style.

Specific comments, with recommendations for addressing each comment

3. Line 22: Please explain the abbreviation PFT

4. Many grammatical errors make it difficult to read the text.

5. Line 43: 'wellknow' \diamond 'well known'

6. I think it is confusing that the authors discriminate between effector-triggered immunity (ETI) and cellular surveillance. Commonly ETI is defined as the sensing of pathogen-mediated disruption of cellular activities (e.g. <https://www.nature.com/articles/s41564-019-0623-2>, <https://www.nature.com/articles/nri3398>). Molecular perturbations that result from a pathogen effector's virulence function, such as inhibition of protein translation, modifications of the actin

cytoskeleton, mitochondrial dysfunction, or changes in ion concentrations, are recognized by host factors (guard proteins) that surveil these core cellular processes. Usually, effector-triggered immunity is opposed to pattern-triggered immunity. The authors distinguish between ETI and cellular surveillance, because “[...], most of the ETI responses still depend on PRRs, [...]” (line 49-50). However, in the paper they refer to (<https://doi.org/10.1371/journal.ppat.1005795>), the term ‘surveillance immunity’ is also used rather as a substitute for effector triggered immunity. I would like to hear the authors’ reasoning for not using the common definition.

7. Line 68: I suggest to include the pathogens that produce the specified toxins

8. Figure 1 legend: Please name the software used for enrichment pathway analyses

9. Line 110: Fig. 1A shows that the term ‘metabolic pathways’, not ‘energy metabolic pathways’. Please adjust the labeling in the figure, or the text.

10. Figure 2A: It is unclear to me what exactly the micrographs show. Please explain, also in the legend of the figure what is shown and indicate interesting and most relevant structures or features by arrows.

11. Figure 2B: How many worms were assessed per treatment? How was the fragmentation ratio determined.

12. Figure 2C: In the figure legend it says: “Analysis of the mitochondrial membrane potential ($\Delta\Psi_m$) under each treatment. The level of $\Delta\Psi_m$ was determined by densitometry of each worm. Around 20 worms were measured in each condition”. The labeling of the y-axis, however, says “Mean density”. Please explain.

13. Line 186-187: Please explain how you did this, e.g. that you used a Rhodamine-labeled Cry5Ba protein

14. Figure 3A: From the micrographs, it is not clear to me if the Cry5Ba protein co-localizes with mitochondria or not. Please elaborate how you came to the conclusion that Cry5B does not co-localize with mitochondria.

15. Figure 3B and C: What is the difference between B and C? Are these two independent runs of the experiments?

16. Line 214-216: Please consider that *aak-2* expression was previously shown to be affected by Bt infection

<https://www.sciencedirect.com/science/article/pii/S0145305X15000336?via=ihub>

17. Figure 4 D and E: How many worms were analyzed?

18. Line 223-224: “Inhibition of mitochondrial fragmentation using *mdivi-1* could significantly suppress the Thr172 phosphorylation of AAK-2 (Fig. 4A).” The difference between infected worm samples and *mdivi-1*-treated worm samples is not really visible, but quantification shows a significant decrease in phosphorylation in *mdivi-1*-treated worm samples. Please provide the two remaining replicates of the Western Blot in a supplementary figure.

19. Line 101 in supplements: Figure S110 \diamond S10

20. Figure 5 C/D: Please label Y-axis consistently with either survival(%) or % alive

21. Figure 5 D and E: Please explain in the figure legend what “Flod of BMB171/Cry5Ba” means.

22. Figures 5 C and 6 B and E: The statistical analysis and test commonly used for survival analysis are Kaplan-Meier analysis followed by log-rank test. Why did the authors use one-way ANOVA for the data shown in Figure 6 B and E and the t-test for the data shown in Figure 5C? Why did the authors not do a Kaplan-Meier analysis followed by log-rank test? How many worms were assessed per replicate?

23. Figure 6D: Which time-point is shown?

24. Line 343-344: Please consider that *sod-3* was previously shown to be required for resistance against the *B. thuringiensis* strain Bt679

<https://journals.plos.org/plospathogens/article?id=10.1371/journal.ppat.1008826>

25. Line 321-322: Why did the authors choose these 8 genes? Are these genes effector genes in the defense against Bt and its PFTs? Has this effector function been demonstrated for these genes? What does ‘typical’ imply in this context?

26. Figure 7 D and F: How many worms were analyzed per replicate and treatment?

27. Figures S13B and S13C are only referred to in the discussion, the results are not presented in the main text.

28. Figure 8 includes MAPK signaling. The findings on this pathway are however only referred to in the discussion, they are not presented in the results section.

Reviewers' comments:

Reviewer #1 (Remarks to the Author):

Ju et al. study *C. elegans* innate immune response to *B. thuringiensis*. The authors concluded that toxins from Bt cause potassium leakage and imbalance of energy via mitochondrial damage. The authors also proposed AMPK-dependent surveillance system as a new strategy for host defense. The study is generally interesting and helpful. However, there are some analysis details missing that would help researchers understand the study.

Response: We appreciate your recognition of the value of our work. We appreciate your numerous helpful suggestions, which greatly assisted us in improving our work. Your thoughtful comments on our statistical analysis helped us revise the corresponding statistical methods. In addition, we re-summarized all of the statistical analyses and included Supplementary Table S5 to show the methodology and outcomes. We refer to the supplemental experiments and research publications as suggested. These suggestions have greatly inspired us. We respond to each of the contents one by one below. Furthermore, we apologize for very loosely characterizing the selected "immune-related genes," so we've outlined their potential activities and provided further information. We have also supplemented and revised other details as indicated below.

Major comments:

1) L321 (Fig. 7A), author indicates p value is obtained from t-test. It seems not possible for application of t-test on overlap of two gene sets. As I know, common tests for such topic should be Fisher exact test or permutation test.

Response: Thanks for your comments on our data statistics. We apologize for employing the incorrect statistical method. We utilized Fisher's exact test to verify the significance of the overlap between different gene sets and labeled in the text (Line 316-317), as advised by you, and the details are presented in Supplementary Table S5.

2) Author declared that worms triggered AMPK to defend against BT infection.

It will be better if author could perform additional gene set enrichment analysis against AMPK pathways (such as aak-2 study from <http://www.ncbi.nlm.nih.gov/pubmed/25723162>, <http://www.ncbi.nlm.nih.gov/pubmed/21331044> and <http://www.ncbi.nlm.nih.gov/pubmed/26959186>).

New Figure.5a. In comparison to previously described gene clusters up-regulated by AAK-2 activation, we discovered that 305 genes up-regulated by Bt infection are likewise AAK-2 targets. The Phenotype Enrichment Analysis according to the Wormbase database revealed that some of these genes are involved in defensive response.

Response: Thank you for your valuable advice. We've included some relevant analytical work. Using prior work by Mair et al. and Burkewitz et al., we gathered a total of 1250 genes previously identified as AAK-2 targets. By comparing these genes to our transcriptome study, we discovered that 305 genes up-regulated by Bt infection are also targets of AAK-2, as shown in Figure 5a, and the relevant genes are presented in the new Supplementary Table S3. The Wormbase database's Phenotype Enrichment Analysis revealed that some of these genes are involved in defensive responses. This investigation demonstrates that AMPK plays an important role during Bt infection, particularly in the response to Bt infection. Please check the revised text Lines 244-249 for a description of this outcome.

3) L351 (Fig. 7B) author selected 8 genes from intersection between BT induced genes and daf-16 targets and declared as “immune-related genes”. However, there is no evidence supporting these genes are immune related (especially for sod-3).

Response: Thank you for your constructively critical comments. We apologize for referring to these chosen genes as immune-related genes in a broad sense. *Thn-2*¹, *lys-7*², *clec-166*^{3,4} and *sod-3*^{2,5,6} were identified as immune-related genes among the 8 genes we chose. The gene *tre-4*⁷, which encodes trehalase, was chosen because it has been related to environmental resistance, while *F32A5.4*⁸ was related to aspartyl proteases, which have been associated with pathogen infection. Furthermore, we chose the most significant upregulated gene, *ttr-44*^{9,10}, which intrigued us because its function is unclear. The last gene is D1086.3, a randomly chosen gene with an unclear function that we inaccurately labeled as an immune-related gene. In this case, we eliminated the data associated with D1086.3 in Figure 7b. In addition, we updated the description of

"immune-related genes" to include more information on these genes. Please check the revision in manuscript Line 317-319 and new Fig. 7b.

Specifically for *sod-3*, it has been demonstrated that *sod-3* is a widely utilized marker to validate the activation of DAF-16^{1,11}. Furthermore, a recent study by Zárates-Potes et al. confirmed that SOD-3 was implicated in *C. elegans*' defense response, particularly to Bt⁶. As a result, SOD-3 was chosen as a target gene. Our supplementary data (new Figure 7e) also proves that the five genes, including *sod-3*, are indeed involved in the defense against BMB171/Cry5Ba. Please check the revision in manuscript Line 342-355, 447-456 and new Fig. 7e.

4) Author confirmed that potassium can reverse the mitochondrial damage caused by BT toxin. Is it also involved in host defense against Bt?

New Figure. 3g. A survival assay reveals that worms' resistance to Bt is enhanced in a potassium-free environment. High potassium, on the other hand, increases worm sensitivity to Bt at low toxin concentrations.

Response: Thank you for suggesting this vital experiment, it was helpful for us to demonstrate that potassium leakage can cause defense against Bt. We discovered that lower quantities of environmental potassium ions can promote worm resistance to Bt, whereas high concentrations had the reverse effect by treating worms with varying concentrations of BMB171/Cry5Ba under 0, 20, and 100mM potassium environments. This scenario is most likely the result of a more robust immune response triggered by significant potassium leakage. While high potassium ion concentrations restore mitochondrial pressure, they also suppress immune response activation. Our findings show that potassium can repair mitochondrial damage, which was implicated in host defense against Bt. The results of the supplemental experiment were inserted as a new Figure 3g. Please check the revisions in manuscript Line 273-276 and new Figure 3g.

Minor:

1) Fig. 7A for 263 *daf-16* target genes. Author should mention that gene set is from class 1 genes in Murphy et al since Murphy et al provide two gene sets in their study.

Response: Thank you for this suggestion. We changed our description to emphasize that this gene set came from class 1 genes in the work of Murphy et al. Please check the revise in manuscript Line 314-315.

2) Fig 6B. Author should indicate how many worms used in the survival assay.

Response: Thank you for your suggestion. In the figure legend of each experiment, we have described the number of worms utilized. When conducting survival assay experiments, we reuse 30-50 worms in every three repeats respectively.

Reviewer #2 (Remarks to the Author):

Brief summary of the manuscript

Ju et al. study the response of *C. elegans* to infection with the *Bacillus thuringiensis* (Bt) strain BMB171/Cry5Ba, which produces the pore forming toxin Cry5Ba. They demonstrate that infection leads to mitochondrial damage and an increased AMP/ATP ratio. Also, the authors show that Bt infection, and in particular exposure to the Cry5Ba toxin, leads to a decrease in potassium ion concentration. Since restoration of intracellular potassium reduced mitochondrial damage and restored the AMP/ATP ratio, the authors conclude that the leakage of potassium caused by exposure to the PFT Cry5Ba leads to mitochondrial damage and subsequently energy imbalance. Furthermore, the authors demonstrate that the energy sensor AMP-activated protein kinase (AMPK) is activated by BMB171/Cry5Ba infection downstream of mitochondrial damage and energy imbalance. The *aak-2(ok524)* mutation that affects the subunit $\alpha 2$ of AMPK, a *aak-2* null mutation, and RNAi silencing of *aak-2* reduced worm survival upon BMB171/Cry5Ba infection and the authors show that *aak-2* functions in the intestine to regulate resistance. Finally, the authors provide evidence of DAF-16-mediated transcriptional activation of gene expression downstream of *aak-2*.

Overall impression of the work

The findings presented in this manuscript are potentially significant and of broad interest. The authors identify *aak-2* as novel regulator of the *C. elegans* response to *B. thuringiensis* PFTs and link *aak-2* to DAF-16-dependent signaling. Moreover, they may have identified a novel potential mechanism of effector-triggered immunity (ETI) in *C. elegans*. Their results suggest that *aak-2* acts as ‘guard protein’ that surveils the energy status of the cell to sense pathogen-mediated mitochondrial damage. The importance of effector-triggered immunity, which is mediated by the surveillance system and not by pattern recognition receptors, for animal defense responses to pathogen attack was established rather recently and our understanding on the role of guard proteins and the processes that they guard in metazoans is still emerging. This paper is thus very timely. Response: Thank you for your appreciation of the value of our work and your helpful suggestions. Based on your suggestions, we have extensively revised the manuscript and supplemented several experiments.

We are very appreciative of the related works recommended by you, which we found motivating. Concerning related studies, we have added several relevant experiments and illustrated associated features in the sections below. In short, we discovered that the phenotype of compound metformin stimulates the energy imbalance disturbed by pathogenic bacteria, and that the specific effect factor regulated by AAK-2 could be involved in resistance to BMB171/Cry5Ba.

Your comments on our statistical analysis are quite useful in revising the corresponding statistical methods. In addition, we re-summarized all of the statistical analyses and showed the methods and results in Supplementary Table S5.

We apologize for the many grammatical errors in the manuscript. We performed major text edits to make the content simpler to read.

We have amended and supplemented your thorough suggestions one by one, and the details are presented below. Thank you once more for your valuable suggestions, which have helped us strengthen our work.

However, to demonstrate that *aak-2* functions in ETI, the authors would need to show that *aak-2* is required to activate immune defenses against BMB171/Cry5Ba. They demonstrate that *aak-2* is required for increased survival after infection, but is this really the consequence of an activated pathogen defense response? What is unclear to me is what constitutes the defense response to BMB171/Cry5Ba infection, e.g. which genes are part of the defense response? The authors make first steps into that direction by investigating the expression of DAF-16 target genes (figure 7B) in the *aak-2* mutant background, but they do not explain the role of these genes in pathogen defense (or why they were chosen). They also investigate activation of the superoxide dismutase gene *sod-3*, but it remains unclear if it is required for defense against BMB171/Cry5Ba and the authors neglect to discuss its known function in defense against another *B. thuringiensis* strain (see comment #24 below). In line 426-427 of the discussion the authors state that “[...] transcription of certain DAF-16-dependent innate immune effectors, which helps *C. elegans* defend against pathogen infection.” The claim of the immune effector function of the analyzed genes is not supported by the experiments done. It is thus unclear if *aak-2* really acts in ETI/surveillance immunity.

Response: We appreciate the reviewer's comments and suggestions. We understand the reviewers' concerns about whether AAK-2 is involved in the response against to Cry5Ba. We demonstrated that AAK-2 activated downstream DAF-16 and p38-MAPK immune pathways. The FOXO transcription factor and p38-MAPK signaling pathway respond directly to Bt and toxin, especially Cry5Ba, and thereby contribute to immune defense¹²¹³. Furthermore, we discovered that the lethal activity of BMB171/Cry5Ba was enhanced in both *aak-2* and *daf-16* mutant worms. As the reviewer proposed, we haven't identified the specific components involved in defense response. So we conducted further experiments and made major improvements to address this issue.

New Figure. 7e. The survival assay of the N2 worms after RNAi knockdown of key genes under BMB171/Cry5Ba treatment.

We agree with the reviewer's concerns about which gene is involved in the defense response against BMB171/Cry5Ba. We used RNAi to knock down seven selected genes (see reply 25 for details on these genes), and then we evaluate the susceptibility of these RNAi worms to BMB171/Cry5Ba (new Figure 7e). Our results reveal that the five genes include *lys-7*, *F32A5.4*, *thn-2*, *sod-3* and *tre-4* all constitute the defense response to BMB171/Cry5Ba, which encode lysozyme and thaumatin, as well as a parasite aspartyl protease inhibitor and effectors such as superoxide dismutase (SOD) and trehalase. We appreciate the review's highly useful suggestions for *sod-3*. So we included previous work about *sod-3*. At the same time, our survival experiments also show that SOD-3 can participate in the defense against Bt infection.

Finally, we apologize for simply defining the selected genes that help *C. elegans* defend against pathogen infection. We hope that our new results and our experimental design ideas can dispel your concerns about AAK-2's participation in ETI. According to our new results, we also revised the paragraph in our discussion "[...] transcription of certain DAF-16-dependent innate immune effectors, which helps *C. elegans* defend against pathogen infection.", and addressed the possible means of *C. elegans* defend BMB171/Cry5Ba. Please check the revisions in manuscript Line 342-355, 447-456 and new Figure 7e.

Another ETI mechanism in metazoans, including *C. elegans*, is surveillance of protein translation. Chemical inhibition of translation induces pathogen defenses in the absence of a pathogen. It would thus be very interesting to test if a decrease in AMP/ATP ratio in the absence of a pathogen induces pathogen defenses and this would also support the idea of *aak-2* mediated ETI. Did the authors consider this?

Added Figure. S9. The compound: metformin and AICAR could significantly increase the AMP/ATP level.

New Figure. 4a. The compound: metformin and AICAR could significantly cause the phosphorylation of AAK-2.

New Figure. S14. The compound: metformin and AICAR could significantly increase DAF-16 nuclear localization without Bt infection.

Response: As the reviewer suggested, we also agree it is was a good idea to use compounds to artificially simulate the infection process of pathogens. So we used a compound: metformin, a drug that has been shown to increase the concentration of intracellular AMP¹⁴. First, we verified that metformin indeed significantly increased the intracellular AMP / ATP level. Further, we found that metformin could activate AMPK using western blot. Finally, we demonstrated that metformin could activate the DAF-16 pathway by affecting the nuclear localization of DAF-16. Through these

results, we can prove that *aak-2* does activate immunity by sensing intracellular energy states caused by Bt infection. We've given a detailed explanation of these findings. Please check these revisions in manuscript Line 218-221, Line 337-340 and Figure 4a, b and S8, 9.

In addition, I have two major concerns:

1. Statistical analysis:

While the experiments were apparently done in a convincing and thorough manner (e.g. they authors used several *aak-2* mutant alleles, RNAi, and performed tissue-specific rescue experiments to demonstrate a role of *aak-2* in survival upon BMB171/Cry5Ba infection) and seem to be technically sound, I am concerned about the statistical analyses of the data. The only statistical tests the authors use to analyze their data are the t-test and one-way ANOVA, both parametric tests. In the case of survival analysis, both tests are not suitable (see comment #22 below) and with the same experimental design they use the t-test in one case and one-way ANOVA in another case (data shown in figure 5C and 6B). Did the authors check if their data are really all normally distributed? Also, the authors mainly present pooled data (i.e. the mean and SD of three independent replicates are shown). They should provide data and results of statistical tests (p-values from all experiments) for each experimental run in a supplementary table.

Response: We appreciate the reviewer's constructive criticism of our data statistical analysis, and we apologize for not performing statistical analysis as suggested, which were excellent. We have unified all the statistics for each experimental and show the details of processes and results in added Supplementary Table S5.

In our survival assay, we counted the survival rate of worms after treatment by different dosages of BMB171/Cry5Ba for 4 days according to published methods¹⁵. Therefore, it isn't applicable for us to use Kaplan-Meier analysis followed by log-rank test. So we performed the statistical analysis on the survival rate at each Cry5Ba concentration. As in previous work showed, we think this method is adequate for checking the susceptibility of worms defend Bt.

2. Writing:

The manuscript is not clearly written. It contains many grammatical errors and is difficult to read. The manuscript requires revision for style.

Response: Thank you for your suggestion. We have involved a native English speaker for language corrections. Specifically, we double-checked and fixed all grammatical problems in the manuscripts. Furthermore, we have made considerable revisions to our complicated and unintelligible explanations, making our manuscript more understandable. We considered submitting a manuscript with all modifications tracked, but we discovered that it would render the manuscript unreadable. As a result, we only highlighted key text modifications in the content of the discussion points below. We hope that you will find our manuscript easier to read.

Specific comments, with recommendations for addressing each comment

3. Line 22: Please explain the abbreviation PFT.

Response: Thanks for your suggestion, we have added a description of PFT. Please check this revise in manuscript Line 21-22.

4. Many grammatical errors make it difficult to read the text.

Response: We apologize for these grammatical errors, We have rechecked our manuscript and revised all errors as we can find. We have also involved the native English speaker in language corrections.

5. Line 43: ‘wellknow’ □ ‘well known’

Response: Thank you for spotting our mistake, We rechecked our manuscript and revised all errors that we discovered. Please check this revision in manuscript Line 42.

6. I think it is confusing that the authors discriminate between effector-triggered immunity (ETI) and cellular surveillance. Commonly ETI is defined as the sensing of pathogen-mediated disruption of cellular activities (e.g. <https://www.nature.com/articles/s41564-019-0623-2>, <https://www.nature.com/articles/nri3398>). Molecular perturbations that result from a pathogen effector’s virulence function, such as inhibition of protein translation, modifications of the actin cytoskeleton, mitochondrial dysfunction, or changes in ion concentrations, are recognized by host factors (guard proteins) that surveil these core cellular processes. Usually, effector-triggered immunity is opposed to pattern-triggered immunity. The authors distinguish between ETI and cellular surveillance, because “[...], most of the ETI responses still depend on PRRs, [...]” (line 49-50). However, in the paper, they refer to (<https://doi.org/10.1371/journal.ppat.1005795>), the term ‘surveillance immunity’ is also used rather as a substitute for effector triggered immunity. I would like to hear the authors’ reasoning for not using the common definition.

Response: Thank you so much for your comments. We also take note of these publications, which encouraged us to think about our work in new ways, and we completely agree with you. We believe that surveillance immunity is a critical component of ETI.

Since Melo et al., proved that *C. elegans* can detect pathogens by monitoring the destruction of the core cell process, subsequent studies have proved that *C. elegans* can identify pathogens by sensing the inhibition of host translation activities, perturbation of mitochondrial homeostasis, disruption of the ubiquitin-proteasome system, DNA damage and so on. The common feature of these mechanisms is that neither the participation of PRRs nor the identification of the toxins is required. Instead, intracellular physiological changes disturbed by toxins are key to cellular surveillance. Therefore, we use ‘surveillance immunity’ to describe ETI responses in *C. elegans*.

We apologize for not conveying our views clearly, thus we revised the description of the ETI response in the introduction section. We emphatically describe that

“surveillance immunity” is an important extension and supplement of ETI response, and state the reason why this is more common in *C. elegans*. On this basis, we want to prove that host can identify pathogens and activate immune responses by monitoring the energy imbalance caused by pathogens. Please check this revision in manuscript Line 47-52 and 380-383.

7. Line 68: I suggest to include the pathogens that produce the specified toxins

Response: Thank you for your suggestion. We have added the description of the relevant pathogen (Line 67-68).

8. Figure 1 legend: Please name the software used for enrichment pathway analyses

Response: Thank you for your suggestion. The GO and KEGG pathway enrichment analysis was performed by the KOBAS database (<http://kobas.cbi.pku.edu.cn/>)¹⁶. We have added the description in the method of “RNA-sequencing and transcriptome analysis” in Figure 1a legend. Please check this revision in manuscript Line 107 and 954.

9. Line 110: Fig. 1A shows that the term ‘metabolic pathways’, not ‘energy metabolic pathways’. Please adjust the labeling in the figure, or the text.

Response: Thank you for spotting our mistake. To maintain the consistency of the enrich analysis, we changed "energy metabolic pathways" in the text to "metabolic pathways" and revised our description. Please check this revision in manuscript Line 109-110.

10. Figure 2A: It is unclear to me what exactly the micrographs show. Please explain, also in the legend of the figure what is shown and indicate interesting and most relevant structures or features by arrows.

Response: Thank you for your suggestion. We marked the different states of mitochondria with triangles and asterisks and as indicated in the figure legends. In addition, we have enlarged the mitochondria in various states as indicated. Please check this revision in new Figure 2a and Line 962-964.

11. Figure 2B: How many worms were assessed per treatment? How was the fragmentation ratio determined.

Response: In this experiment, we define the mitochondria fragmentation (MF) phenotype when more than 50% of the mitochondria in the worm are punctate. we randomly selected roughly 50 worms from each condition to observe their mitochondrial morphology by fluorescence microscope and counted the number of MF worms. We have added a description of judgment criteria in the experimental method. Please check this revision in Line 516-518.

12. Figure 2C: In the figure legend it says: “Analysis of the mitochondrial membrane potential ($\Delta\Psi_m$) under each treatment. The level of $\Delta\Psi_m$ was determined by

densitometry of each worm. Around 20 worms were measured in each condition". The labeling of the y-axis, however, says "Mean density". Please explain.

Response: Thank you for your comments. To quantify the mitochondrial membrane potential, we converted the fluorescence intensity of each worm to the optical density value and used the cumulative optical density/ worm area as the quantitative value of membrane potential intensity. As you pointed out, "Mean density" used in the figure is not suitable, so we change it to "Mean fluorescence intensity/ worm ($\Delta\Psi_m$)" in figure 2c; figure 3f refers to a similar method¹⁷.

13. Line 186-187: Please explain how you did this, e.g. that you used a Rhodamine-labeled Cry5Ba protein

Response: Thank you for your suggestion. In this experiment, worms were labeled with endocytosis protein RT311[Pvha-6::GFP::RAB-11] and mitochondria SJ4143(zcIs17 [Pges-1::GFPmt]) were fed by the Rhodamine-labeled Cry5Ba protein, respectively. The fluorescence channels for worms (green) and Cry5Ba (red) were displayed separately and then merged. We have revised figure 3a and indicated the details in the manuscript (Line 177-180, 977-980).

14. Figure 3A: From the micrographs, it is not clear to me if the Cry5Ba protein co-localizes with mitochondria or not. Please elaborate how you came to the conclusion that Cry5B does not co-localize with mitochondria.

Response: Thank you for your comments. Please refer to reply 13 for further experiment details. According to the picture, we can observe the yellow signal of Cry5Ba colocalized with endocytic vesicle, which means the colocalization of Cry5Ba with endocytotic processes. However, Cry5Ba and mitochondria show their separate red and green signals which indicates Cry5B does not co-localize with mitochondria. We have revised figure 3a and a description of the judgment criteria has been included in the figure legend. Please check these revisions in new Figure 3a and Line 977-980.

15. Figure 3B and C: What is the difference between B and C? Are these two independent runs of the experiments?

Response: Thank you for your comments. These two results are carried out concurrently. There is no distinction between them. We split it into two figures considering the layout. Now, we have merged them to new figure 3b.

16. Line 214-216: Please consider that aak-2 expression was previously shown to be affected by Bt infection <https://www.sciencedirect.com/science/article/pii/S0145305X15000336?via=ihub>

Response: Thank you for your suggestion. Yang et al. compared the transcriptome and proteome of *C. elegans* under Bt infection. Two potential immunomodulators, CRH-1 and AAK-2, were discovered to be linked with adenosine monophosphate (AMP). We agree with you that this work contributes to our point of view, and we have cited it in the amended manuscript (Line 205-206).

17. Figure 4 D and E: How many worms were analyzed?

Response: Thank you for your comments. We have described how many worms were utilized in each experiment. In Figure 4d, we counted the mitochondrial morphology of 40-50 worms. In Figure 4e, we collected 1000 - 2000 worms treated by Bt for nucleotide extraction and LC/ MS analysis.

18. Line 223-224: "Inhibition of mitochondrial fragmentation using mdivi-1 could significantly suppress the Thr172 phosphorylation of AAK-2 (Fig. 4A)." The difference between infected worm samples and mdivi-1-treated worm samples is not really visible, but quantification shows a significant decrease in phosphorylation in mdivi-1-treated worm samples. Please provide the two remaining replicates of the Western Blot in a supplementary figure.

Response: Thank you for your suggestion. As you suggested, we used metformin to increase the cellular ratio of AMP / ATP. After being treated with metformin, we discovered that AAK-2 was considerably phosphorylated. Furthermore, we re-examined the phosphorylation of the protein AAK-2 when exposed to BMB171/Cry5Ba in different potassium environments. Thus, we have replaced the Western blot figure with a new figure 2a and 2c, and we also put the duplicate results in supplementary figure S8.

We found that mdivi-1 influenced the activation of AMPK to a certain extent but not completely. We suspect that mdivi-1 cannot completely inhibit AMP levels. However, we still observe that the phosphorylation level of AAK-2 has been inhibited. We think this is enough to support our view that mdivi-1 inhibits AMPK activation by inhibiting mitochondrial fragmentation.

19. Line 101 in supplements: Figure S110 □ S10

Response: Thank you for spotting our mistake. We have corrected the figure annotations in our manuscript and Supplementary Material.

20. Figure 5 C/D: Please label Y-axis consistently with either survival(%) or % alive

Response: Thank you for spotting our mistake. We have unified Y-axis to "survival(%)" in figure 3g, 5d, 5e, 6d and 6e.

21. Figure 5 D and E: Please explain in the figure legend what "Flod of BMB171/Cry5Ba" means.

Response: Thank you for your comments. In our work, we found that Cry5Ba toxin was the primary cause of mitochondrial damage. As a result, we quantified the concentration of Cry5Ba in the survival experiment, and subsequently, BMB171 /Cry5Ba with different dilution ratios were employed to treat worms. After four days, the survival rates were calculated concurrently. The concentration of Cry5Ba is used here to illustrate distinct gradient dilutions of BMB171/Cry5Ba. To clearly illustrate our experiment, we change this to "Concentration of Cry5Ba" in each figure.

22. Figures 5 C and 6 B and E: The statistical analysis and test commonly used for survival analysis are Kaplan-Meier analysis followed by log-rank test. Why did the authors use one-way ANOVA for the data shown in Figure 6 B and E and the t-test for the data shown in Figure 5C? Why did the authors not do a Kaplan-Meier analysis followed by log-rank test? How many worms were assessed per replicate?

Response: Thank you very much for your effective suggestion on our data statistics. According to published methods¹⁵, we treated worms with different concentrations of BMB171 /Cry5Ba for 4 days, and then counted the survival rate in each concentration. Therefore, Kaplan-Meier analysis is not suitable for our statistical analysis in Figure 5c,5f and 6e. To solve this, we use the student's t-test to analyze each pair of survival data (see in reply 1). The details of processes and results have been shown in added Supplementary Table 5. However, for Figure 6b, we have re-implemented the data analysis by Kaplan-Meier analysis followed by log-rank test, the details have also been shown in added Supplementary Table 5.

23. Figure 6D: Which time-point is shown?

Response: Thank you for your comments. We counted the survival rates of several rescue worms after Bt infection for 96 hours. In the legend, we included a description of the time-point (Line 1037-1038). In addition, we have provided a processing time explanation to each appropriate experiment legend.

24. Line 343-344: Please consider that *sod-3* was previously shown to be required for resistance against the *B. thuringiensis* strain Bt679 <https://journals.plos.org/plospathogens/article?id=10.1371/journal.ppat.1008826>

Response: Thanks for your suggestion. *sod-3* has been mentioned to be involved in *C. elegans* against pathogens previously. Zárate-Potes et al. demonstrated that the effector gene *sod-3* is specifically required for defense against Bt infection. This article is still important for us to indicate the correlation of *sod-3* as an immune factor, and we have cited it. Our new data shows it also participated in the defense against Bt. Please check in the revised manuscript Line 317 to 319, 342-355 and 447-456.

25. Line 321-322: Why did the authors choose these 8 genes? Are these genes effector genes in the defense against Bt and its PFTs? Has this effector function been demonstrated for these genes? What does 'typical' imply in this context?

Response: Thank you for your constructive critical comments. These genes were chosen by comparing DAF-16 regulatory targets to our transcriptome data. Among the overlapping genes in Table S3, we can only find these genes which have been reported to be related to the defense response.

Among the 8 genes we selected, *thn-2*¹, *lys -7*², *clec-166*^{3,4} and *sod-3*^{2,5,6} were identified as immune-related genes, The gene *tre-4*⁷ encoding trehalase was selected because it has been related to environmental resistance; F32A5.4⁸ was related to aspartyl proteases which have been shown to be involved in pathogen infection. Furthermore, we selected the most significant upregulated gene *ttr-44*^{9,10}, which

intrigued us because its function is unknown. The last gene is a randomly selected gene *D1086.3* with an unknown function, which we had mistakenly defined as an immune-related gene. Thus we deleted the data associated with this incorrectly selected gene in Figure 7b.

We apologize for loosely defining these selected genes as typical immune-related genes. So we revised the description of “immune-related genes” to include more information on these genes. Please check the revise in manuscript Line 317-319 and new Fig. 7b

26. Figure 7 D and F: How many worms were analyzed per replicate and treatment?

Response: Thank you for your comments. We have described the number of worms used in each experiment in the figure legend. In figure 7d, at least 30 worms had been analyzed in each treatment, and 20 worms for 7f (new figure 7g). In addition, we remeasured the fluorescence intensity of each worm and included the data in revised figure 7f.

27. Figures S13B and S13C are only referred to in the discussion, the results are not presented in the main text.

Response: Thank you for your comments. We have not carried out further studies on p38-MAPK pathways, so we simply included these findings in the discussion section of the initial manuscript. Now, we have included a section in the revised text to describe these findings since they help us prove that *aak-2* stimulates the immune response. Please check these revises in Line 356-371.

28. Figure 8 includes MAPK signaling. The findings on this pathway are however only referred to in the discussion, they are not presented in the results section.

Response: Thank you for your comments. We have added a section to describe these results. See in reply 27 for the details.

Reference

- 1 Evans, E. A., Kawli, T. & Tan, M.-W. *Pseudomonas aeruginosa* suppresses host immunity by activating the DAF-2 insulin-like signaling pathway in *Caenorhabditis elegans*. *PLoS Pathog.* **4**, e1000175 (2008).
- 2 Zhou, M., Liu, X., Yu, H. & Gong, J. Lactobacillus Regulates *Caenorhabditis elegans* Cell Signaling to Combat *Salmonella* Infection. *Front Immunol* **12**, 653205-653205 (2021).
- 3 Liu, J., Hafting, J., Critchley, A. T., Banskota, A. H. & Prithiviraj, B. Components of the cultivated red seaweed *Chondrus crispus* enhance the immune response of *Caenorhabditis elegans* to *Pseudomonas aeruginosa* through the pmk-1, daf-2/daf-16, and skn-1 pathways. *Appl. Environ. Microbiol.* **79**, 7343-7350, (2013).
- 4 Pees, B. *et al.* Effector and regulator: Diverse functions of *C. elegans* C-type lectin-like domain proteins. *PLoS Pathog* **17**, e1009454, (2021).
- 5 Zou, C. G., Tu, Q., Niu, J., Ji, X. L. & Zhang, K. Q. The DAF-16/FOXO transcription factor functions as a regulator of epidermal innate immunity. *PLoS Pathog* **9**, e1003660 (2013).
- 6 Zarate-Potes, A. *et al.* The *C. elegans* GATA transcription factor elt-2 mediates distinct transcriptional responses and opposite infection outcomes towards different *Bacillus thuringiensis* strains. *PLoS Pathog* **16**, e1008826, (2020).
- 7 Pellerone, F. I. *et al.* Trehalose metabolism genes in *Caenorhabditis elegans* and filarial nematodes. *Int J Parasitol* **33**, 1195-1206 (2003).
- 8 Brown, A., Girod, N., Billett, E. E. & Pritchard, D. I. Necator americanus (human hookworm) aspartyl proteinases and digestion of skin macromolecules during skin penetration. *Am. J. Trop. Med. Hyg.* **60**, 840-847 (1999).
- 9 Tepper, R. G. *et al.* PQM-1 complements DAF-16 as a key transcriptional regulator of DAF-2-mediated development and longevity. *Cell* **154**, 676-690, (2013).
- 10 Wang, X. *et al.* *Caenorhabditis elegans* transthyretin-like protein TTR-52 mediates recognition of apoptotic cells by the CED-1 phagocyte receptor. *Nat Cell Biol* **12**, (2010).
- 11 Chávez, V., Mohri-Shiomi, A., Maadani, A., Vega, L. A. & Garsin, D. A. Oxidative stress enzymes are required for DAF-16-mediated immunity due to generation of reactive oxygen species by *Caenorhabditis elegans*. *Genetics* **176**, 1567-1577, (2007).
- 12 Wang, J., Nakad, R. & Schulenburg, H. Activation of the *Caenorhabditis elegans* FOXO family transcription factor DAF-16 by pathogenic *Bacillus thuringiensis*. *Dev Comp Immunol* **37**, 193-201 (2012).
- 13 Kao, C. Y. *et al.* Global functional analyses of cellular responses to pore-forming toxins. *PLoS Pathog* **7**, e1001314 (2011).
- 14 Zhang, L., He, H. & Balschi, J. A. Metformin and phenformin activate AMP-activated protein kinase in the heart by increasing cytosolic AMP concentration. *Am. J. Physiol. Heart Circ. Physiol.* **293**, H457-466, (2007).
- 15 Bischof, L. J., Huffman, D. L. & Aroian, R. V. Assays for toxicity studies in *C. elegans* with Bt crystal proteins. *Methods Mol Biol* **351**, 139-154 (2006).
- 16 Xie, C. *et al.* KOBAS 2.0: a web server for annotation and identification of

- enriched pathways and diseases. *Nucleic Acids Res* **39**, W316-322, (2011).
- 17 Egge, N. *et al.* Trauma-induced regulation of VHP-1 modulates the cellular response to mechanical stress. *Nat Commun* **12**, 1484 (2021).

Reviewers' comments:

Reviewer #1 (Remarks to the Author):

The authors addressed most of my concerns. Only one minor suggestion.

1) Figure 5A, author used Phenotype Enrichment Analysis according to Wormbase database to generate this figure but not cite the tools.

Reviewer #2 (Remarks to the Author):

The authors have made considerable and commendable efforts to address the various criticisms and suggestions made by reviewers. The resulting manuscript is improved especially with respect to the role of an increased AMP/ATP level in triggering an immune response and the function of AAK-2 in regulating defenses to Cry toxins.

However, my concern regarding the presentation of the data is still not sufficiently addressed. The authors added Supplementary table 5, which provide the results of the statistical test, but because it only shows the p values it is still unclear if data was pooled and if yes from how many experimental runs ('repeats'). The information is in part given in the materials and methods section, in part in the figure legends, but it is absolutely necessary to specify the number of experiments and if data are pooled or representative in the figure legend for all graphs. What do the three data points (white circles) represent in figures 1b, 2b, d, and e, 3 d and e (and so on). Are these the mean values of three experimental runs? Please clarify.

In supplementary table 5 it is sometimes unclear which groups were compared and it would be helpful to state the comparison (X vs. Y) in an additional column for each p-value (the authors already did this for some of the comparisons in the column 'description'). It is also custom in the field to provide not only the results of statistical analyses of the pooled data, but the results of the statistical tests with p-values of each performed experimental run, which is important to get an idea of between-replication variability and/or the raw data in supplementary files.

Minor suggestions:

I suggest to change the title to "C. elegans monitors energy status via the AMPK pathway to trigger innate immune responses against bacterial pathogens"

Line 17: first sentence is unclear, in particular "[...] triggering pattern of host innate immune system[...]". What is meant by pattern of host innate immune system?

Line 118-119: Please mention (for the potentially non-expert reader) that the bre-5 mutant is resistant to toxin exposure.

Line 372: Please state the 6 'typical' p38 MAPK genes also here in the main text.

Please mention in the legend of Figure 1b how the AMP/ATP ratio was measured.

Please change MF to mitochondria fragmentation in the legend of Figure 2. Line 977-978: Are "mitochondria in the split state" equal to mitochondria showing fragmentation? If yes, please say so. Line 980: Please clarify what is meant by "different strains". Please mention that bre-5(RNAi) worms are resistant to toxin exposure. Please explain the labels "FCCP" and "mdivi-1" also in the figure legend. Please indicate how many experimental runs you did. For example, Figure 2b, 50 worms were counted, are they all from one experimental run? I see three data points (white circles), what do they represent? Means from three independent runs with 50 worms, each? From looking at TableS5 it seems that the data are all from one run or pooled. Please clarify for EACH figure (not only figure 2). Please clarify in the description of how mitochondrial membrane potential was measured (2c) where

the fluorescence is coming from. You should also add in the material and methods section what you have written in your letter "To quantify the mitochondrial membrane potential, we converted the fluorescence intensity of each worm to the optical density value and used the cumulative optical density/ worm area as the quantitative value of membrane potential intensity".

Figure 3: Please indicate in the legend how potassium concentration was measured. Please clarify for each figure how many independent experimental runs were done and what the three data points (white circles in graphs) represent.

Figure 4d: Please explain the labels "AICR", "mdivi-1", and "pL4440" (empty vector RNAi control) also in the figure legend. The reader should be able to understand the figures without having to check the main text or materials and methods.

Figure 5: Please indicate for the x-axis in figure 5b what is meant by 0, 1/8, 1/4, 1/2 and 1. 5C: How many experimental runs ("repeats") were done? What is shown, a representative run or pooled data? Please indicate. If pooled data is shown, please provide p-values from all experimental runs ("repeats") in a supplementary file.

Figure 6b: Please indicate which statistical test you used to analyze the data. In your letter you state that you did Kaplan-Meier analysis followed by log-rank test.

Reviewers' comments:

Reviewer #1 (Remarks to the Author):

The authors addressed most of my concerns. Only one minor suggestion.

1) Figure 5A, author used Phenotype Enrichment Analysis according to Wormbase database to generate this figure but not cite the tools.

Response: Thank you for your valuable advice. We have cited related work published by David Angeles-Albores et al. in the revised manuscript Line 247.

Reviewer #2 (Remarks to the Author):

The authors have made considerable and commendable efforts to address the various criticisms and suggestions made by reviewers. The resulting manuscript is improved especially with respect to the role of an increased AMP/ATP level in triggering an immune response and the function of AAK-2 in regulating defenses to Cry toxins.

However, my concern regarding the presentation of the data is still not sufficiently addressed. The authors added Supplementary table 5, which provide the results of the statistical test, but because it only shows the p values it is still unclear if data was pooled and if yes from how many experimental runs ('repeats'). The information is in part given in the materials and methods section, in part in the figure legends, but it is absolutely necessary to specify the number of experiments and if data are pooled or representative in the figure legend for all graphs. What do the three data points (white circles) represent in figures 1b, 2b, d, and e, 3 d and e (and so on). Are these the mean values of three experimental runs? Please clarify.

Response: Thank you for your suggestion on the presentation of the data. We apologize for leaving out some details while describing the experimental procedures and state the data, your comments tremendously aided us in presenting the results. All legends now include the number of experimental repeats and whether data were pooled. Besides, Supplementary Table 5 now includes this information as well. Our independent three data points reflect the mean values of three experiments, so we add text descriptions in the figure legends to clarify the meaning of data points in each figure. We have highlighted all changes in the manuscript text file

In supplementary table 5 it is sometimes unclear which groups were compared and it would be helpful to state the comparison (X vs. Y) in an additional column for each p-value (the authors already did this for some of the comparisons in the column 'description'). It is also custom in the field to provide not only the results of statistical analyses of the pooled data, but the results of the statistical tests with p-values of each performed experimental run, which is important to get an idea of between-replication variability and/or the raw data in supplementary files.

Response: Thank you for your suggestion on the presentation of the data again. Your

advice is very timely and effective. Now we have added the description of which groups were compared in Supplementary Table 5, as well as the experiment repeats and whether the data pooled. In figure legends, we have also supplemented the meaning of data points, sample size and experimental repeats. Your suggestion about extra statistical tests with p-values of each performed experimental run is very good, which may help to show more details. Since we averaged the replicates of each experiment repeat, and we emphasize the differences between groups. So we showed the averages of independent replicates in the figure to assist readers estimate the variability of replicates, the related data have been shown in raw data. This, as far as we know, is likewise a standard display strategy to show the desired result. We hope our supplementary description addresses your concerns about the statistics.

Minor suggestions:

I suggest to change the title to “*C. elegans* monitors energy status via the AMPK pathway to trigger innate immune responses against bacterial pathogens”

Response: Thank you for your suggestion. We have revised the title.

Line 17: first sentence is unclear, in particular “[...] triggering pattern of host innate immune system[...]”. What is meant by pattern of host innate immune system?

Response: Thank you for your comment. We have modified this sentence to “Pathogen recognition and the triggering of host innate immune system are critical to understanding pathogen-host interaction.” Please check this revision in manuscript Line 17-18.

Line 118-119: Please mention (for the potentially non-expert reader) that the *bre-5* mutant is resistant to toxin exposure.

Response: Thank you for your suggestion. We have added the description that *bre-5* mutant worms is resistant to toxin exposure. Please check this revision in manuscript Line 115-117.

Line 372: Please state the 6 ‘typical’ p38 MAPK genes also here in the main text.

Response: Thank you for your suggestion. We have stated the 6 typical genes *pmk-1*, *tir-1*, *nsy-1*, *sek-1*, *kgb-1* and *jun-1* in the text. Please check this revision in manuscript Line 365.

Please mention in the legend of Figure 1b how the AMP/ATP ratio was measured.

Response: Thank you for your suggestion. We have stated the AMP/ATP ratio was measured by LC/MS. Please check this revision in manuscript Line 964.

Please change MF to mitochondria fragmentation in the legend of Figure 2. Line 977-

978: Are “mitochondria in the split state” equal to mitochondria showing fragmentation? If yes, please say so. Line 980: Please clarify what is meant by “different strains”. Please mention that *bre-5*(RNAi) worms are resistant to toxin exposure. Please explain the labels “FCCP” and “mdivi-1” also in the figure legend. Please indicate how many experimental runs you did. For example, Figure 2b, 50 worms were counted, are they all from one experimental run? I see three data points (white circles), what do they represent? Means from three independent runs with 50 worms, each? From looking at TableS5 it seems that the data are all from one run or pooled. Please clarify for EACH figure (not only figure 2).

Response: Thank you for your valuable suggestions. The figure legend has been extensively altered to make it easier to understand. We have added a supplementary description of MF. Besides, we revised the description of “mitochondria in the split state”, indicating it is equal to mitochondria fragmentation. We have clarified that “different strains” represent different Bt treatments. We added the description of *bre-5* (RNAi) worms. We have also added a description for the utilized compound FCCP, mdivi-1, etc. At last, we have added the experiment repeats and the meaning of the data points in each figure. We have highlighted all changes in the manuscript text. Please check these revisions in manuscript Line 974 - 987.

Please clarify in the description of how mitochondrial membrane potential was measured (2c) where the fluorescence is coming from. You should also add in the material and methods section what you have written in your letter “To quantify the mitochondrial membrane potential, we converted the fluorescence intensity of each worm to the optical density value and used the cumulative optical density/ worm area as the quantitative value of membrane potential intensity”.

Response: Thank you for your valuable suggestions. We have added a description of the mitochondrial membrane potential measurement method in the legend of Fig. 2b, notably referencing the use of the fluorescent dye tetramethyl rhodamine ethyl ester. This procedure for membrane potential data analysis also has been added to the methods section. Please check these revisions in manuscript Line 531-535 and 978-980.

Figure 3: Please indicate in the legend how potassium concentration was measured. Please clarify for each figure how many independent experimental runs were done and what the three data points (white circles in graphs) represent.

Response: Thank you for your valuable suggestions. We have added a description of the potassium concentration measurement method in the figure legend, notably referencing the use of the fluorescent dye ION Potassium Green-2 AM. We also clarified the number of experiment repeats, as well as the meaning of the data points in the figures. Please check these revisions in manuscript Line 997-998 and highlight texts.

Figure 4d: Please explain the labels “AICR”, “mdivi-1”, and “pL4440” (empty vector RNAi control) also in the figure legend. The reader should be able to understand the figures without having to check the main text or materials and methods.

Response: Thank you for your valuable suggestions. We have added a description of the used compounds and the meaning of the plasmid vector. Please check these revisions in manuscript Line 1030-1032.

Figure 5: Please indicate for the x-axis in figure 5b what is meant by 0, 1/8, 1/4, 1/2 and 1. 5C: How many experimental runs (“repeats”) were done? What is shown, a representative run or pooled data? Please indicate. If pooled data is shown, please provide p-values from all experimental runs (“repeats”) in a supplementary file.

Response: Thank you for your suggestions. The “0, 1/8, 1/4, 1/2 and 1” represent the different concentrations of BMB171/Cry5Ba, up to 7.4 µg/ml Cry5Ba. We have revised the figure and added a description of the meaning. For all experiments, we have added the description of experimental repeats and the meaning of the data point. Experimental repeats have been also added to Supplementary Table 5. Please check these revisions in manuscript Line 1041-1042 and highlight revisions.

Figure 6b: Please indicate which statistical test you used to analyze the data. In your letter you state that you did Kaplan-Meier analysis followed by log-rank test.

Response: Thank you for your suggestions. We have added the description of the statistical method of Kaplan-Meier analysis followed by log-rank test. Please check these revisions in manuscript Line 1061-1062 and figure 6b.